# T1: Advancing Language Model Reasoning through Reinforcement Learning and Inference Scaling

Zhenyu Hou [1]  Xin Lv [2]  Rui Lu [1]  Jiajie Zhang [1]  Yujiang Li [1]  Zijun Yao [1]  Juanzi Li [1]  Jie Tang [1]  Yuxiao Dong [1]

## Abstract

Large language models (LLMs) have demonstrated remarkable capabilities in complex reasoning tasks. However, existing approaches mainly rely on imitation learning and struggle to achieve effective test-time scaling. While reinforcement learning (RL) holds promise for enabling self-exploration, recent attempts yield modest improvements in complex reasoning. In this paper, we present T1 to scale RL by encouraging exploration and understand inference scaling. We first initialize the LLM using synthesized chain-of-thought data that integrates trial-and-error and self-verification. To scale RL training, we promote increased sampling diversity through over-sampling. We demonstrate that T1 with open LLMs as its base exhibits inference scaling behavior and achieves superior performance on challenging math reasoning benchmarks. More importantly, we present a simple strategy to examine inference scaling, where increased inference budgets directly lead to T1's better performance without any additional verification. The model weights and training data are publicly available at https://github.com/THUDM/T1.

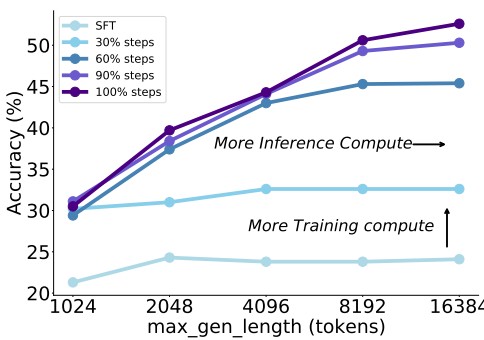

Figure 1: Training and inference scaling of T1 (Qwen2.5-32B) on AIME2024 with different max generation budgets ($x$-axis) and different RL training steps (SFT: 0% RL steps). T1 achieves better performance with increasing generation length. More RL training steps bring more significant inference scaling. Details in Section 4.

## 1. Introduction

Large language models (LLMs) (Achiam et al., 2023; Team et al., 2023; Dubey et al., 2024) have recently exhibited remarkable capabilities in addressing complex reasoning tasks (Shao et al., 2024; Lozhkov et al., 2024; Zhu et al., 2024; Zhou et al., 2024). The chain-of-thought (CoT) paradigm (Wei et al., 2022) has been instrumental in enhancing LLM reasoning, emphasizing the importance of constructing and refining reasoning paths (Zelikman et al., 2022; Gulcehre et al., 2023). Recent approaches priori-

tize the imitation learning stage, with efforts dedicated to generating reasoning paths through prompting (Yu et al., 2024; Mitra et al., 2024; Yue et al., 2024) or rejection sampling (Yuan et al., 2023), followed by training LLMs to replicate the selected reasoning processes.

Despite these advancements, reinforcement learning (RL)—which can enable LLMs to self-explore and learn from feedback—has demonstrated greater potential than imitation learning (OpenAI, 2024) for unlocking inference scaling, i.e., test-time scaling of LLMs. However, its development within the research community remains limited to date. Previous studies (Shao et al., 2024; Wang et al., 2024; Hou et al., 2024) suggest that RL yields relatively modest performance improvements in complex reasoning and lacks scalability compared to its earlier training stages.

Regarding test-time scaling, existing methods typically rely on repeated sampling (Brown et al., 2024), where multiple outputs are generated from a given policy model and auxiliary verifiers (Snell et al., 2024) are used to select the best response. However, these approaches do not update to the policy model itself, thus failing to fundamentally improve the reasoning ability of LLMs. Repeatedly sampling short responses with verifiers also falls short of the expected inference scaling behavior (OpenAI, 2024). Ideally, deeper thinking and longer generation are expected

[1]Tsinghua University [2]ZhipuAI. Correspondence to: Yuxiao Dong <yuxiaod@tsinghua.edu.cn>.

*Proceedings of the 42nd International Conference on Machine Learning*, Vancouver, Canada. PMLR 267, 2025. Copyright 2025 by the author(s).

to directly lead to better performance without relying on external signals. Consequently, improving LLM reasoning through RL scaling and inference scaling remains an underexplored challenge.

In this work, we explore the key factors for scaling RL and enabling test-time scaling behavior in LLM reasoning. We introduce T1, which exhibits superior reasoning capabilities trained via RL. We demonstrate its promising inference scaling behavior in reasoning tasks, as shown in Figure 1. The core idea of T1 is to encourage extensive exploration in RL to scale its training while applying appropriate penalties to maintain training stability.

First, we finetune the LLM using synthesized CoT data with trial-and-error and self-verification, which helps substantially expand the exploration space before RL training. This strategy makes it different from previous works (An et al., 2023; Yuan et al., 2023; Zhang et al., 2024) that typically focus on the correct steps but usually overlook the overall thinking process.

Next, to scale RL, we promote greater sampling diversity during RL training by oversampling responses for each prompt with a high temperature. In addition, we adopt a token-level entropy bonus and an on-policy KL normalization strategy to encourage varied token generation. These also help mitigate excessive regularization from the reference. Strict penalties are imposed on repetitive or nonsensical outputs to prevent collapse and stabilize RL training.

Third, building on this scaled RL training, we propose a simple way to measure and understand inference scaling. The idea is to explicitly separate the generation of intermediate reasoning steps from the final answer. This allows us to manually control the inference budget by truncating the reasoning process and study how inference cost affects LLMs' reasoning performance. Figure 1 illustrates the training and inference scaling behavior of the policy model on AIME2024. With the same amount of inference budget, the performance improves consistently as RL training scales. T1 also demonstrates stronger inference scaling trends as training steps increase. In contrast, the T1-SFT and early-stage policy models (e.g., 30% RL steps) show marginal improvements even with the max inference budgets.

We build T1 on top of open models such as Qwen (Yang et al., 2024a) and GLM (GLM et al., 2024). These non-o1 style models are equipped with long thinking through T1's RL scaling, without relying on directly-distilled long Chain-of-Thought data. We evaluate the models on college- and competition-level math reasoning benchmarks. Experiments show that the T1 models achieve superior performance across all benchmarks. For example, T1 with Qwen-32B as its base can outperform the recent Qwen QwQ-32B-Preview model on MATH500, AIME2024, and Omni-MATH-500.

More importantly, T1 exhibits promising trends in both training and inference scaling. The model weights and the data for SFT and RL training are publicly available at https://github.com/THUDM/T1.

## 2. Building T1 with RL Scaling

### 2.1. Preliminary

**Supervised Fine-Tuning (SFT).** In the initial phase of alignment, the pre-trained model is fine-tuned to replicate high-quality demonstration data (e.g., dialogue, summarization). This process, commonly referred to as SFT, serves as a foundational step for aligning the model's outputs with human-like performance.

**Reinforcement Learning from Human Feedback (RLHF).** To further align the fine-tuned model $\pi_\theta$ with human preferences, Ouyang et al. (2022) proposes the use of RL to maximize a reward signal while regularizing the model $\pi_\theta$, that is, to optimize the objective:

$$J_r(\pi_\theta) = \mathbb{E}_{\boldsymbol{x} \sim p_{\text{data}}, \boldsymbol{y} \sim \pi_\theta} \left[ r(\boldsymbol{x}, \boldsymbol{y}) - \beta \log \frac{\pi_\theta(\boldsymbol{y} \mid \boldsymbol{x})}{\pi_{\text{ref}}(\boldsymbol{y} \mid \boldsymbol{x})} \right] \tag{1}$$

Here, $r(\cdot)$ represents the reward function, which evaluates the quality or correctness of each response. It takes a prompt $\boldsymbol{x}$ and its corresponding response $\boldsymbol{y}$ as input and produces a scalar reward. The term $\pi_{\text{ref}}$ refers to the reference model, typically the SFT model.

The general RLHF pipeline proceeds as follows: given a prompt $\boldsymbol{x}$, the policy model $\pi_\theta$ generates $K$ different responses, denoted as $(\boldsymbol{y}_1, \ldots, \boldsymbol{y}_K)$. The reward function then assigns a scalar reward to each pair $(\boldsymbol{x}, \boldsymbol{y}_i)$. Subsequently, the policy model $\pi_\theta$ is updated via reinforcement learning to maximize the objective defined in Eq. 1.

### 2.2. Scaling Reinforcement Learning for Reasoning

We present T1 to scaling RL with the goal of advancing the reasoning capability of LLMs. The core idea behind T1 is to promote exploration during RL training. To achieve this, we propose to expand the search space of the LLM and encourage diverse reasoning trajectories while maintaining the training stability with proper penalties to stably scale the RL training.

The first step of T1 is to initialize the SFT model with rich reasoning patterns, such as trial-and-error and verification—elements often overlooked in prior works as they tend to emphasize (only) the correct steps while neglecting the underlying thought process. The second step is to develop strategies to effectively scale RL training that further contributes to the improvement of LLM reasoning. Figure 2 illustrates the overall framework of T1.

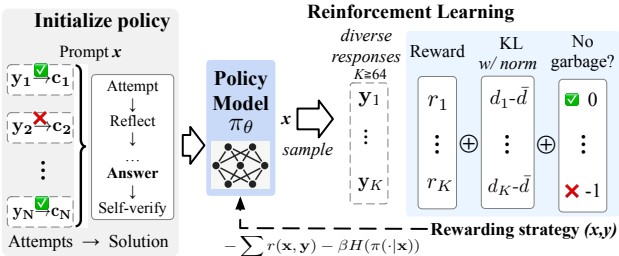

Figure 2: The overall pipeline of RL and T1.

### 2.2.1. INITIALIZING POLICY W/ COT

The chain-of-thought (CoT) (Wei et al., 2022) often defines the reasoning paths of LLMs (OpenAI, 2024), influencing the response search space and RL scaling. To encourage exploration and broaden the search space, we first initialize the policy model with diverse reasoning patterns in the form of CoT (e.g., reflection and verification) in the SFT stage. In doing so, the SFT model is expected to produce responses that incorporate self-correction from flawed attempts as well as thoroughly-verified approaches. The policy model is then used for subsequent RL training. Note that previous works (Zelikman et al., 2022; An et al., 2023) that optimize CoT often undervalue trial-and-error processes, focusing primarily on correct reasoning steps.

Specifically, we begin by generating multiple responses from different LLMs as *attempts* $(\boldsymbol{y}_1, ..., \boldsymbol{y}_N)$ for a given prompt $\boldsymbol{x}$ and judge their correctness based on its ground-truth label. Then we prompt an LLM to thoroughly examine each attempt to obtain the critic $c_i$, including 1) identifying the nature of the errors for incorrect attempts and reflecting on their underlying causes, and 2) performing a verification process for correct ones to confirm the validity of the conclusions derived.

To obtain the reasoning path, we further prompt an LLM to incorporate these refined attempts $\{\boldsymbol{x}, \boldsymbol{y}_i, c_i\}_{i=1}^{N}$—both corrected misconceptions and validated reasoning—into a single output. This is used to illuminate the trial-and-error process that can lead from flawed initial attempts to the final correct solution. Furthermore, we find that some constructed CoTs could simply enumerate different approaches and finally present the correct solution. To overcome this issue, we ask an LLM to rewrite the CoT based on the abstracted pattern to obtain the final solution for SFT training.

### 2.2.2. ENCOURAGING EXPLORATION IN RL TRAINING

To scale RL training, we introduce strategies to encourage exploration. For generation, we adopt a hard sampling strategy to promote the policy model to explore as many trajectories as possible to reach the correct path. For optimization, we integrate the response entropy bonus as an auxiliary loss and also ease the KL regularization to facilitate scaling.

**Scaling response sampling with high temperature.** Scaling sampling aims to capture a broad spectrum of reasoning paths by generating an increased number of responses per prompt during RL training. As indicated in previous work (Li et al., 2024a), the policy model has possessed strong ability in its inherent sampling space, and it is crucial to search extensively for the valuable reasoning path that can enable effective learning for the policy model.

Formally, given a prompt $\boldsymbol{x}$, we sample $K$ responses and obtain $\mathcal{D} = \{(\boldsymbol{x}, \boldsymbol{y}_1), (\boldsymbol{x}, \boldsymbol{y}_2), \ldots, (\boldsymbol{x}, \boldsymbol{y}_K)\}$. We find that a larger $K$ works better in practice and thus adopt $K = 64$ in general. Previous works (Touvron et al., 2023; Kazemnejad et al., 2024; Hou et al., 2024) commonly adopt $K \leq 8$ except for $K = 32$ in Qwen2.5-Math (Yang et al., 2024b).

To further improve response diversity, we utilize a high temperature $\tau$ during sampling. A higher temperature $\tau > 1$ flattens the probability distribution, increasing the likelihood of sampling less probable tokens, and thus encouraging the exploration of diverse token sequences. This strategy facilitates the generation of various reasoning paths and avoids falling into a fixed pattern. Our experiments demonstrate that high sampling temperatures can help training stability and improve the performance gainss in RL training.

For optimization, we use the leave-one-out strategy used in RLOO (Ahmadian et al., 2024) to normalize the rewards:

$$\bar{r}_i = r_i - \frac{1}{k-1}\sum_{j\neq i}^{K} r_j \tag{2}$$

where $r_i$ denotes the reward from reward models or the correctness in $\{0, 1\}$. Consequently, the policy model can learn from a richer set of experiences, leading to improved generalization and performance on reasoning.

**Auxiliary entropy bonus.** To encourage LLMs to generate diverse tokens and avoid deterministic response patterns, we incorporate an entropy bonus into the RL loss function. The modified loss function $\mathcal{L}$ is defined as:

$$\mathcal{L} = \mathcal{L}_{\text{RL}} - \alpha H(\pi(\cdot|\boldsymbol{x}))$$
$$= \mathcal{L}_{\text{RL}} + \alpha \sum_{j}^{|\boldsymbol{y}|} \sum_{w\in\mathcal{V}} \pi_\theta(w|\boldsymbol{x}, \boldsymbol{y}_{:j-1}) \log \pi_\theta(w|\boldsymbol{x}, \boldsymbol{y}_{:j-1}) \tag{3}$$

where $\mathcal{L}_{\text{RL}}$ represents the standard RL loss, $\alpha$ is a weighting coefficient, $H(\pi(\cdot|\boldsymbol{x}))$ is the token-level entropy given the prompt $\boldsymbol{x}$, $\mathcal{V}$ is the vocabulary. The entropy term measures the uncertainty in the token generation process, incentivizing the model to explore tokens of lower probability.

**On-policy KL normalization.** In RL training, the Kullback-Leibler (KL) divergence is used to force the policy model to remain close to the reference model and thus

prevent forgetting and reward hacking (Ouyang et al., 2022). However, a fixed reference anchor could hinder the reward optimization and thus prevent RL scaling. We adopt two strategies to overcome this problem.

First, similar to the reward normalization which scales the rewards to have zero mean, we enforce KL normalization by subtracting the average KL divergence within responses from a prompt $x$, effectively keeping the final normalized reward centered at zero. Denoting the KL for a prompt-response pair $(\boldsymbol{x}, \boldsymbol{y}_i)$ as $d_i = \sum_j^{|\boldsymbol{y}_i|} \log \frac{\pi_\theta(y_{i,j}|\boldsymbol{x})}{\pi_{\text{ref}}(y_{i,j}|\boldsymbol{x})}$, the normalized KL is:

$$\bar{d}_i = d_i - \frac{1}{k-1}\sum_{j \neq i}^{K} d_j \tag{4}$$

Additionally, we apply the Exponential Moving Average (EMA) to dynamically update the reference model, thus avoiding it lagging too behind the policy model:

$$\theta_{\text{ref}}(t) = \alpha\theta_{\text{ref}}(t-1) + (1-\alpha)\theta(t)$$

where $\alpha$ is the decay rate. The EMA provides a smoothed estimate of the policy parameters, serving as a stable reference. With these two strategies, policy updates are incremental and controlled, preventing too large shifts that could destabilize the model training.

### 2.2.3. PENALIZING UNEXPECTED PATTERNS

Encouraging exploration is essential for discovering effective reasoning strategies. However, it is also crucial to discourage unexpected and undesirable response patterns. This helps prevent training collapse and keeps the model from deviating in unintended directions. To achieve this, we implement a straightforward penalty mechanism by assigning a negative reward of $-1$ to responses exhibiting common issues such as repetition, overlong text, and garbage text, e.g., mixed multilingual content or garbled characters.

Specifically, the reward function $r'$ is adjusted as follows:

$$r' = \begin{cases} -1 & \text{if a bad pattern is detected in } \boldsymbol{y}, \\ r & \text{otherwise.} \end{cases}$$

where $r$ is the original reward and $\boldsymbol{y}$ is the generated response. This formulation penalizes responses that exhibit undesirable patterns, guiding the LLM to avoid the following behaviors during training:

- **Repetition and Overlong Text**: Responses containing repetitive n-grams or exceeding predefined maximum length will receive a $-1$ reward. This discourages the generation of meaningless verbose and repeated outputs.

- **Garbage Text**: Responses containing mixed languages or garbled characters are also penalized with a $-1$ reward,

identified through language detection and text quality assessment. This prevents the generation of incoherent or unreadable output. During training, we observe that the policy model can produce fluent but semantically irregular text, leading to a significant increase in response entropy. To address this, we implement both rule-based detection of low-quality text and perplexity-based filtering to enhance training stability.

This penalty effectively prevents the training process from collapsing and steers the model away from generating outputs that could undermine the reasoning capabilities.

## 3. Evaluating T1

We build T1 by using open models, including GLM-4-9B (GLM et al., 2024), Qwen2.5-14B, and Qwen2.5-32B (Yang et al., 2024a). We evaluate the performance on widely-used math reasoning benchmarks—AIME, Omni-MATH (Gao et al., 2024), MATH (Hendrycks et al., 2021), and GPQA (Rein et al., 2023). Accuracy (Pass@1) is used as the primary evaluation metric. The details of the experimental setup is listed in Appendix A.

Table 1: Experiment results on challenging reasoning benchmarks. We report the Accuracy(%) for all datasets.

| | MATH500 | AIME | Omni-MATH-500 | GPQA |
|---|---|---|---|---|
| GPT-4o | 76.6 | 9.3 | 26.8 | 53.6 |
| Claude-3.5-sonnet | 78.3 | 16.0 | / | 65.0 |
| Llama-3.3-70B-Instruct | 73.9 | 24.2 | 27.9 | 50.5 |
| Qwen2.5-Math-7B-Instruct | 82.7 | 16.7 | 29.7 | 36.9 |
| o1-preview | 85.5 | 44.6 | / | 72.3 |
| QwQ-32B-preview | 90.6 | 50.0 | 46.6 | 58.2 |
| GLM-4-9B-chat | 50.1 | 1.7 | 12.9 | 30.9 |
| T1-SFT (GLM-4-9B) | 60.2 | 4.1 | 20.0 | 37.2 |
| T1 (GLM-4-9B) | 65.8 | 9.2 | 24.4 | 38.1 |
| Qwen2.5-14B-Instruct | 78.9 | 13.7 | 30.1 | 45.5 |
| T1-SFT (Qwen2.5-14B) | 77.2 | 10.3 | 28.5 | 42.3 |
| T1 (Qwen2.5-14B) | 87.4 | 30.5 | 38.6 | 48.3 |
| Qwen2.5-32B-Instruct | 82.8 | 13.6 | 33.1 | 49.5 |
| T1-SFT (Qwen2.5-32B) | 83.4 | 24.9 | 34.6 | 49.5 |
| T1 (Qwen2.5-32B) | 92.4 | 50.6 | 49.6 | 56.1 |

### 3.1. Performance Results

Table 1 shows the overall results on major math benchmarks. It is observed that the T1-SFT (without RL scaling) models show promising performance advantages over their original counterparts, respectively. Further with scaling RL, T1 helps achieve significant performance gains over both SFT and original baselines. Specifically, T1 (Qwen2.5-32B) trained with RL achieves over a 10% improvement on Omni-

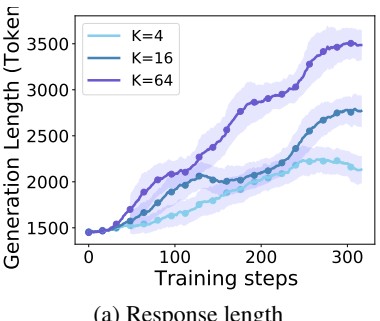

(a) Response length

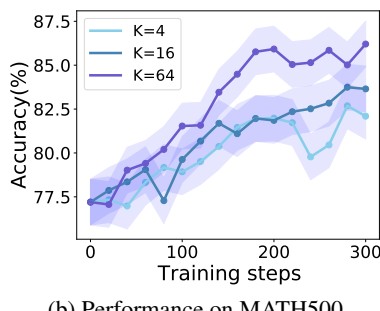

(b) Performance on MATH500

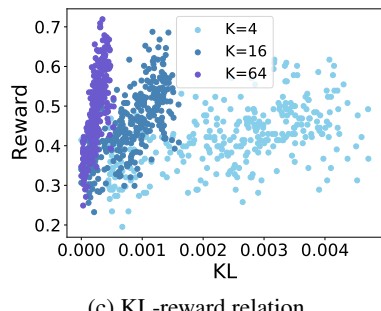

(c) KL-reward relation

Figure 3: Training and evaluation with different number of responses ($K$) sampled in training on T1 (Qwen2.5-14B).

MATH-500 and MATH500 and over a 20% improvement on AIME over its T1-SFT version, showcasing the substantial contribution of RL in reasoning capabilities.

Furthermore, in math-related benchmarks, T1 shows advantages over baselines. For instance, on MATH500, T1 (Qwen2.5-32B) achieves a score of 92.4, outperforming the previous best result (90.6). On AIME, the model also achieves super competitive results compared to baselines. These results highlight the superiority of our approach and demonstrate its effectiveness in boosting the ability to handle complex mathematical reasoning tasks.

While our method is primarily optimized for math-related tasks, it also exhibits out-of-domain (OOD) performance improvements on the GPQA benchmark, where no task-specific optimization is applied in the training. But we also observe remarkable performance improvement in these two benchmarks, indicating that the learned reasoning capability can be generalized across different tasks.

### 3.2. The Effect of Encouraging Exploration

**Sampling more responses encourages exploration.** Figure 3 shows the effect of sampling different number of responses (i.e., $K$) during RL training. We have the following observations: First, sampling more responses during RL training significantly boosts exploration, leading to a substantial increase in response length, as shown in Figure 3 (a). Second, for a fixed number of prompts, more sampled responses accelerate performance improvements, with models achieving better accuracy on tasks like MATH500 in fewer training steps, as evident in Figure 3 (b). Additionally, models trained with more responses yield higher rewards for the same KL divergence and exhibit slower KL growth, as seen in Figure 3 (c). This indicates a better trade-off between KL and reward.

Figure 4 illustrates the consistent improvements with sampling more responses in RL training. It is also observed that sampling additional responses significantly enhances generalization. Although our training data predominantly

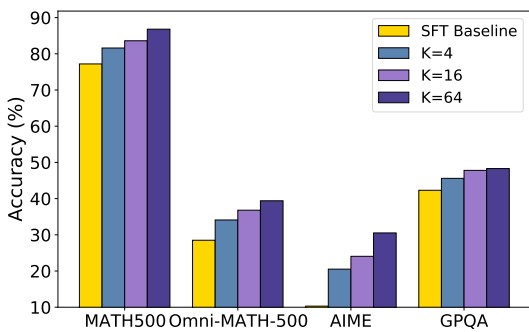

Figure 4: Evaluation results of T1 (Qwen2.5-14B) using different numbers of sampled responses ($K$) in RL training.

Table 2: Effects of sampling parameters in RL training on T1 (Qwen2.5-14B). We use top-$p$=0.95 for all experiments.

| Temperature | min-$p$ | MATH500 | AIME | Omni-MATH-500 |
|---|---|---|---|---|
| 0.9 | 0 | 78.2 | 19.1 | 32.0 |
| 1.1 | 0 | 84.6 | 29.0 | 37.8 |
| 1.2 | 0 | 86.4 | 29.3 | 38.6 |
| 1.3 | 0 | 84.6 | 24.3 | 36.4 |
| 1.2 | 0.05 | 78.8 | 11.5 | 31.6 |

consists of mathematical content and includes almost no science-related data (i.e., physics, chemistry, or biology), T1 demonstrates notable performance improvements in GPQA. The impact of response sampling is particularly pronounced when increasing the number of samples to 64, producing a substantial improvement (over 6%), while sampling only 4 responses shows little to no benefit (around 3%).

**High temperature in sampling benefits RL training.** A higher sampling temperature can encourage the policy model to generate more diverse responses, but RL training is often sensitive to the chosen temperature. Table 2 presents the results of using different sampling temperatures. We observe that higher temperatures contribute to more stable training while training with a temperature $\leq 1.0$ often collapses after just a few steps. This behavior might occur because, at lower temperatures, if the model starts produc-

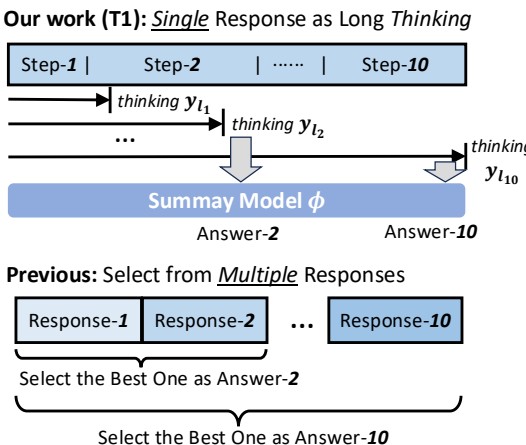

Figure 5: Inference scaling in T1 (Top) and repeated sampling in previous works (Down).

ing undesirable patterns—such as repetitive or nonsensical outputs—it becomes increasingly difficult for it to sample a coherent response, preventing recovery from such issues.

Furthermore, training with a moderately high temperature (e.g., 1.2) yields better performance, but excessively high temperatures also negatively impacts outcomes. Although previous studies (Nguyen et al., 2024) have proposed using min-$p$ sampling to mitigate the risks associated with higher temperatures, our experiments find that this approach can make the training process more prone to collapse, especially exacerbating repetition during generation. Therefore, we only use top-$p$ with high temperature in our training.

Table 3: Effects of penalty reward during RL training on T1 (Qwen2.5-14B). It shows the overlong ratio and accuracy on MATH500. "OverLongRatio" denotes the ratio of generated responses exceeding the configured maximum length.

|  | Penalty | step40 | step80 | step120 | step160 |
|---|---|---|---|---|---|
| OverLongRatio | ✓ | 0% | 2.6% | 1.6% | 0.7% |
|  | ✗ | 0% | 4.1% | 16.3% | - |
| Accuracy(%) | ✓ | 78.6 | 80.1 | 81.2 | 81.2 |
|  | ✗ | 79.0 | 79.2 | 76.4 | - |

**Effects of penalty.** Table 3 demonstrates the impact of penalizing unexpected patterns during training. Without penalty, we observe that the generation length becomes unstable and grows explosively after approximately 100 steps, leading to substantial deterioration in performance. In contrast, the model quickly stabilizes when the penalty is applied, although there is a minor increase in the overlong ratio during the early training stages. The penalized model maintains appropriate generation lengths and produces text with minimal repetition and noise, leading to consistent performance improvement.

## 4. Understanding Inference Scaling in T1

With T1, we aim to improve our understanding of inference scaling, which describes how LLMs achieve performance gains from increased compute during inference (Snell et al., 2024; OpenAI, 2024). Unlike previous works (Snell et al., 2024; Brown et al., 2024; Kumar et al., 2024), which mainly focus on scaling through *repeated sampling*, this study investigates a different approach. We explore how a *single longer generation* (long thinking) affects the correctness of responses and LLMs' reasoning ability.

**How to measure inference scaling?** By scaling RL, T1 enables a new perspective for examining inference scaling. Naturally, when we solve a complex problem, the reasoning process often involves a sequence of thinking steps. Even if an intermediate step is wrong, it still plays a key role in figuring out the solution. Through reflection, the wrong step can help reevaluate the approach, refine the reasoning process, and lead to the correct path. That said, each previous step in the (long) sequence is crucial, as the final solution is built upon existing thinking.

Inspired by this assumption, we outline a simple strategy to analyze the single (long) response generated by T1, as illustrated by Figure 5 (Top). The idea is to truncate each response from the beginning to different lengths, although response length can be controlled through specialized prompting or post-RL fine-tuning. This is used to simulate varying scales of inference cost. Specifically, we truncate each T1 response $\boldsymbol{y}$ into various lengths of tokens $\boldsymbol{y}_{:l_i}$, with $l_i$ represents the length of the truncated response. For each of the truncated response $\boldsymbol{y}_{:l_i}$, we propose to use a summarization model $\phi(\boldsymbol{y}_{:l_i}) \rightarrow A_{:l_i}$ to generate the final answer $A_{:l_i}$. In the analysis below, we simplify by setting $l_i$ as $i \times 10\%$ of the original length of $\boldsymbol{y}$, and by using the base model of T1 as the corresponding summarization model. Since the summarization model can produce complete answers regardless of whether the thinking process has fully concluded, we evaluate $A_{:l_i}$ to assess performance on benchmarks.

This simple strategy enables us to analyze the relationship between thinking/reasoning length and performance, that is, the effect of inference scaling. Figure 6 shows the result on AIME, Omni-MATH-500, and MATH500, as $x$-axis representing the average token count of $\boldsymbol{y}_{:l_i}$ for different responses with respect to each $l_i$. On each benchmark, performance improves consistently as the average number of thinking tokens increases, reflecting the positive impact of longer thinking on LLMs' reasoning performance. Take AIME in Figure 6 for example, the summarization model's accuracy steadily rises from 24% to 50% as more and more thinking tokens are generated. These results indicate that scaling the inference process of LLMs by increasing the number of tokens generated leads to significant improvements in reasoning performance. The consistency of this

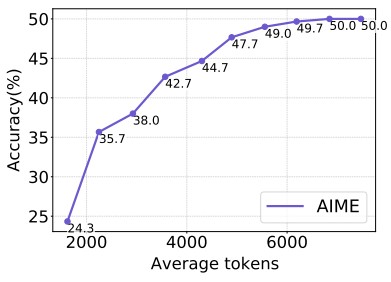 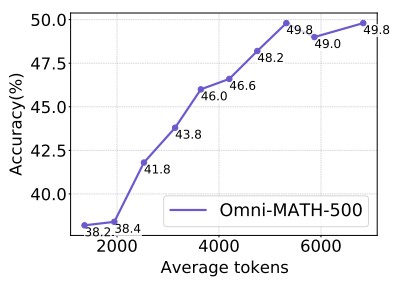 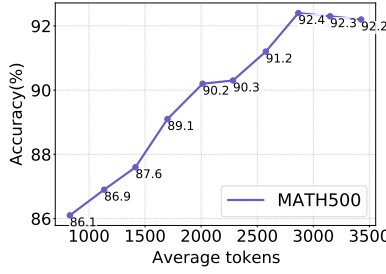

Figure 6: Inference scaling performance of T1 (Qwen2.5-32B) with truncated thinking in AIME, Omni-MATH-500, and MATH500. $x$-axis: the number of thinking tokens used for the summarization model $\phi$ to generate the final answer $A$.

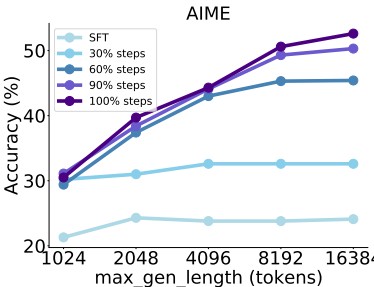 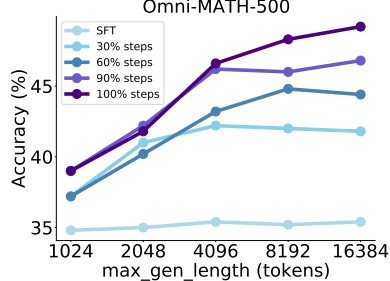 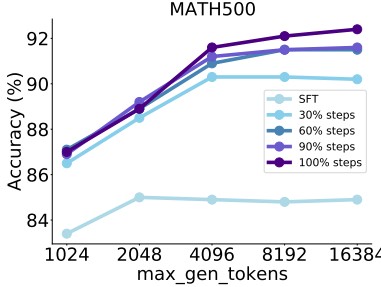

Figure 7: Training and inference scaling of T1 (Qwen2.5-32B) at different RL training steps under different maximum generation budgets. $x$-axis: the maximum allowed length of thinking, and thinking that exceeds the max-length is truncated.

inference scaling effect across different benchmarks suggests that longer inference budgets directly contribute to better performance. This also demonstrates the effectiveness of our simple strategy to measure inference scaling.

**Inference scaling is closely related to RL training scaling.** Based on the above strategy, we further study the relationship between RL training and inference scaling. Figure 7 illustrates the inference scaling behavior of T1 with different RL training compute. The policy model consistently demonstrates improved performance across all three datasets as training compute increases (from bottom to top), under the same maximum generation budget. However, both the T1-SFT model and under-trained RL policy models (e.g., 30% steps in RL) show minimal gains when scaling up inference costs (from left to right). More trained RL models (e.g., 60% steps) achieve significant performance gains. This suggests that more RL training could activate and enhance the inference scaling property. In addition, we can observe that challenging tasks benefit more from inference scaling. For example, with 100% RL training steps (purple lines), T1 gains a 66% relative improvement (from 30% to 50%) on AIME, 30% (from 38% to 49%) on Omni-math-500, and 6% (from 86% to 92%) on MATH500.

We further study the inference behavior during RL training, as shown in Figure 8. First, the policy model is encouraged to produce increasingly longer responses with the presented exploration strategies during training, as seen in Figure 8 (a). Next, we use an LLM to classify each reasoning step into

different patterns:*Correct mistake*, *Try different approach*, *Verification*, and others. This helps study the reasoning behavior in longer responses. Figure 8 (b) shows the count of different reasoning patterns corresponding to the purple line in Figure 8 (a). Longer responses contain more diverse reasoning patterns, suggesting the model engaged in trial-and-error problem-solving. An initial drop in reasoning attempts likely indicates that the model first learned to eliminate redundant steps caused by the SFT. Overall, these results demonstrate that RL training enhances the model's ability to leverage increased inference compute for more effective reasoning.

**Case study.** As described in Section 2.2.1, the T1 model is capable of exploring different approaches and correcting reasoning errors. We aim to identify the *key* reasoning steps—those in which the model discovers the correct idea to solve a problem—and investigate whether these steps share common characteristics.

Since our model can rectify its mistakes, it is not appropriate to judge the correctness of a reasoning step solely based on whether it leads to the correct final result. To address this, we employ a strategy inspired by truncated inference scaling analysis: a step is labeled as a key step when an additional reasoning step transforms the model's summarized answer from incorrect to correct. Using this approach, we identify 126 instances from the Omni-math-500 dataset and 108 instances from the AIME dataset where such transitions occur. We then conduct a frequent word analysis on these

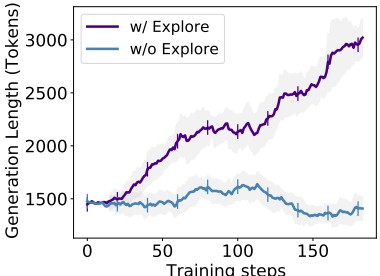

(a) Length *w/* and *w/o* encouraging exploration during RL training.

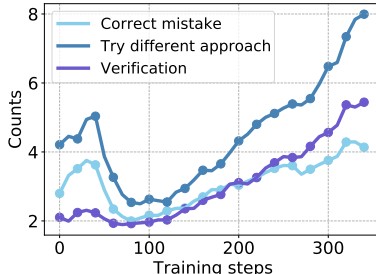

(b) The number of different reasoning patterns during RL training with exploration.

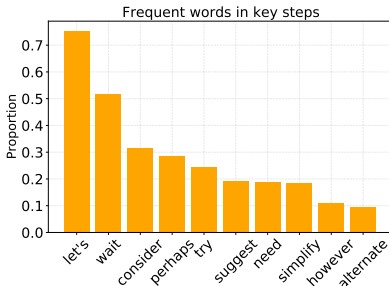

(c) Word frequency distribution in key reasoning steps. *key step* refers to the step that, if omitted, yields an incorrect answer, but when included, leads to the correct result.

Figure 8: A study of the inference behavior of T1 (Qwen2.5-14B) during RL training.

key steps, with the results illustrated in Figure 8 (c). Notably, words such as *wait*, *perhaps*, and *alternate* frequently appear in these steps. These terms often signal moments of rethinking or the exploration of alternative approaches. The observation further demonstrates that reflection is a key capability in improving the reasoning ability.

Figure 9 in Appendix A illustrates an example of the reasoning process of T1. Initially, the model attempts to solve the problem until it reaches an intermediate step. It is not confident in the reasoning and then employs a different approach and arrives at the same result. Finally, the model performs an additional verification step to ensure that no potential cases are overlooked. At this point, it is confident that it has arrived at the correct answer.

## 5. Related Work

**Language model reasoning.** Recent advances in language models in complex reasoning tasks have been remarkable and have shown substantial improvements (Lewkowycz et al., 2022; Shao et al., 2024; Lightman et al.). One line of work involves pretraining large language models (LLMs) on extensive reasoning-related datasets, such as mathematics and code, which has significantly improved their foundational understanding of reasoning tasks (Paster et al., 2023; Shao et al., 2024; Lozhkov et al., 2024). Another line of work focuses on post-training strategies. Some works (Yuan et al., 2023; Yue et al.; Zelikman et al., 2022; Li et al., 2024b) focus on synthesizing reasoning-related question-answer pairs and incorporating additional verifiers or critics to improve data quality Additionally, fine-tuning models through reinforcement learning (Ouyang et al., 2022; Bai et al., 2022) enables the model to learn from feedback and self-guided actions. This iterative process allows models to critique and refine their answers, thus improving their problem-solving abilities(Shao et al., 2024; Wang et al., 2024; Kazemnejad et al., 2024). In this work, we build on the third line of work by scaling RL techniques.

**Scaling language models.** Scaling is one of the key factors leading to the success of powerful LLMs and provides crucial insights into the continuous improvement. Kaplan et al. (2020); Hoffmann et al. (2022); Du et al. (2024) study the scaling laws for pretraining and demonstrate that scaling model size and training tokens can both lead to predictable improvements. Recently, reinforcement learning as well as test-time scaling for LLMs to boost reasoning capabilities has attracted much attention since the emerge of OpenAI o1 (OpenAI, 2024), but are still under-explored in the open community. Gao et al. (2023); Cobbe et al. (2021) explore the scaling laws in reward modeling under a synthetic setting and Rafailov et al. (2024a) studies the scaling of direct policy optimization (Rafailov et al., 2024b). Hou et al. (2024) investigates the impact of scaling and shows that traditional methods are not scalable and are far from effective as shown in o1 in boosting the reasoning abilities of LLMs. Beyond scaling RL training, inference scaling is also a crucial yet under-explored. Existing works (Brown et al., 2024; Snell et al., 2024) measure the inference cost by repeated sampling, which heavily relies on external supervision as a verifier and is not as scalable as proposed in o1.

## 6. Conclusion

In this paper, we present T1 for enhancing large language models' reasoning capabilities through scaled reinforcement learning. By promoting extensive exploration during RL training while maintaining stability through strategic penalties and oversampling, T1 achieves strong reasoning performance and demonstrates promising test-time scaling behavior. We introduce a novel approach to measuring inference scaling by analyzing the relationship between reasoning steps and model performance, revealing that increased RL training improves both reasoning accuracy and inference scaling trends. Experimental results demonstrate that T1 shows excellent performance and outperforms existing models on challenging reasoning benchmarks.

## Impact Statement

This paper advances the field of machine learning by improving language model reasoning capabilities through reinforcement learning and inference scaling. Our work contributes to the development of AI systems with enhanced reasoning abilities. We foresee that our work will have positive impacts on research in language model development and applications. The ethical aspects and societal implications of our work align with those commonly associated with advancing the field of machine learning. There are many potential societal consequences of our work, none which we feel must be specifically highlighted here.

## Acknowledgement

This work has been supported by the Natural Science Foundation of China (NSFC) 62495063, Tsinghua University (Department of Computer Science and Technology) - Siemens Ltd., China Joint Research Center for Industrial Intelligence and Internet of Things (JCIIOT), and the New Cornerstone Science Foundation through the XPLORER PRIZE.

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

# A. Appendix

## A.1. Setup

**Data construction.** The prompts used in the training data all come from publicly available datasets, including MATH-train (Hendrycks et al., 2021), and NuminaMath (Li et al., 2024b). We split around 12k for the SFT stage and the others for RL training. To prepare the data for reinforcement learning, we convert the original instances into *(Question, Label)* pairs through the following two steps: The first step is *answer extraction.* Since NuminaMath provides a combined rationale and final answer, we employ an LLM to extract the answer as the label and shorten it as much as possible. A concise label facilitates more accurate correctness checks by the LLM during later training stages. For MATH-train, we directly use the question and official answer provided in the dataset. The second step is *data filtering.* For both NuminaMath and MATH-train, we further filter the data to ensure reliability and high quality for RL training. In particular, we remove inaccurate (noisy) labels and overly simple problems. For each question, we generate 16 responses and retain only those instances whose pass rate lies in the interval $(0, \delta)$ (where $\delta = 0.3$ in our experiments). Finally, we got 30k data that can be used for RL training.

---

**An example from AIME-2024 Question-15 solved by T1**

Let $\mathcal{B}$ be the set of rectangular boxes with surface area 54 and volume 23. Let $r$ be the radius of the smallest sphere that can contain each of the rectangular boxes that are elements of $\mathcal{B}$. The value of $r^2$ can be written as $\frac{p}{q}$, where $p$ and $q$ are relatively prime positive integers. Find $p + q$.

---

{**Problem Analysis**} Let the dimensions of the box be $a$, $b$, and $c$. From the surface area constraint: $2(ab + bc + ac) = 54 \implies ab + bc + ac = 27$. From the volume constraint: $abc = 23$. Our goal is to minimize $a^2 + b^2 + c^2$ under these constraints.

{**Symmetry Attempt**} Perhaps the box is close to a cube. Let's try $a = b$ as a simpler case. This gives:

$$2(a^2 + 2ac) = 54 \implies a^2 + 2ac = 27 \implies c = \frac{27 - a^2}{2a}$$

Combined with volume constraint $a^2c = 23$, $\cdots$ (*simplification*), we get:

$$a^3 - 27a + 46 = 0$$

{**Numerical Trial**} Testing $a = 2$, $\cdots$ (*verification*), this gives $a = b = 2$ and $c = \frac{23}{4}$.

{**Another Approach**} Let's consider another approach using Lagrange multipliers to minimize $a^2 + b^2 + c^2$ subject to the constraints:

$$L(a, b, c, \lambda, \mu) = a^2 + b^2 + c^2 - \lambda(ab + bc + ac - 27) - \mu(abc - 23)$$

$\cdots$ (*heavy calculation of partial derivatives*). So $a = b$ is an optimal condition.

{**Calculation**} For $a = 2$, $b = 2$, $c = \frac{23}{4}$,

$$4r^2 = a^2 + b^2 + c^2 = 4 + 4 + \frac{529}{16} = \frac{657}{16}$$

{**Verification**} Since $657 = 3^2 \times 73$ and $64 = 2^6$, these numbers are relatively prime. Therefore, $p = 657$ and $q = 64$, giving $p + q = 721$.

{**Try other values**} Wait, let's make sure we've considered all possibilities. Let's try $a = 1$. $\cdots$ (*derive equation and calculate $\Delta$*), but $a = 1$ does not satisfy $a \leq \frac{2}{\sqrt{23}} \approx 0.4226$. Therefore, $a$ cannot be 1.

{**Final Answer**} Answer: $\boxed{721}$

---

Figure 9: An example of T1's response to Question-15 in AIME2024. T1 explores multiple solution approaches and employs various verification strategies to ensure the correctness of the final answer.

**Training details.** For SFT, we train the models for three epochs using a learning rate 1e-5 with cosine decay scheduling. For RL training, we sample 64 responses for each prompt and perform policy gradient descent for every 32 prompts. We train the model with a 1.5e-6 learning rate and KL set to 2e-4. For the reward function, we use the ground truth, i.e., the correctness of the response, as the metric, assigning a reward of 1 for correct answers and 0 for incorrect ones. Although using a trained reward model is generally considered a superior approach due to its ease of optimization, we find that using the correctness of response as the reward also performs well for reasoning tasks and helps mitigate issues such as data

distribution shifts and reward hacking. If not specified, the max generation length for training and inference is set to 10,240 for GLM-4-9B and Qwen2.5-14B models and 16,384 for Qwen2.5-32B models.

**Evaluation.** We evaluate the performance of the models on the test sets of reasoning-related datasets, including AIME2024[1], Omni-MATH (Gao et al., 2024), MATH (Hendrycks et al., 2021), and GPQA (Rein et al., 2023) Accuracy (Pass@1) is used as the primary evaluation metric. We report the results of baselines from previous works if available. For MATH, we assess performance on a subset of the MATH-test set, referred to as MATH500, following the predefined split in Lightman et al.. For Omni-MATH, we sample a smaller evaluation subset by sampling 500 examples called Omni-MATH-500 for efficient yet comprehensive evaluation. GPQA consists of graduate-level problems in biology, physics, and chemistry. For AIME, we use the official questions released for the year 2024, which consists of 30 problems. We evaluate each model 32 times on AIME to get stable results and report the average performance. For all datasets, we use the greedy sampling strategy with SGLANG (Zheng et al., 2023) as the inference engine.

---

**A full example from AIME-2024 Question-15 solved by T1**

**Question**

Let $\mathcal{B}$ be the set of rectangular boxes with surface area 54 and volume 23. Let $r$ be the radius of the smallest sphere that can contain each of the rectangular boxes that are elements of $\mathcal{B}$. The value of $r^2$ can be written as $\frac{p}{q}$, where $p$ and $q$ are relatively prime positive integers. Find $p + q$.

**Response from T1**

Let's tackle this problem step by step. We are given a set of rectangular boxes with surface area 54 and volume 23. Let the dimensions of the box be $a$, $b$, and $c$. The surface area and volume constraints give us:

$$2(ab + bc + ac) = 54 \implies ab + bc + ac = 27$$

$$abc = 23$$

Our goal is to find the radius $r$ of the smallest sphere that can contain each of these boxes. The smallest sphere that can contain a rectangular box is one that contains all the vertices of the box. This means the diameter of the sphere must be at least the space diagonal of the box, which is:

$$d = \sqrt{a^2 + b^2 + c^2}$$

Therefore, the radius $r$ must satisfy:

$$2r \geq \sqrt{a^2 + b^2 + c^2} \implies r \geq \frac{\sqrt{a^2 + b^2 + c^2}}{2}$$

To find the smallest possible $r$, we need to minimize $a^2 + b^2 + c^2$ under the given constraints.
First, let's express $a^2 + b^2 + c^2$ in terms of $a + b + c$, $ab + bc + ac$, and $abc$. Recall the identity:

$$(a + b + c)^2 = a^2 + b^2 + c^2 + 2(ab + bc + ac)$$

Rewriting this, we have:

$$a^2 + b^2 + c^2 = (a + b + c)^2 - 2(ab + bc + ac)$$

We know $ab + bc + ac = 27$ from the surface area constraint. Let $s = a + b + c$. Then:

$$a^2 + b^2 + c^2 = s^2 - 2 \times 27 = s^2 - 54$$

Now, we need to express $s$ in terms of $a$, $b$, and $c$. Since $abc = 23$, let's consider the symmetric sums. We might try to find $s$ directly, but it's not immediately obvious. Perhaps we can use the Arithmetic Mean–Geometric Mean inequality $AM \geq GM$ to find a lower bound for $s$.
Wait, let's apply $AM \geq GM$ to $a$, $b$, and $c$:

---

[1] https://huggingface.co/datasets/AI-MO/aimo-validation-amc

$$\frac{a+b+c}{3} \geq \sqrt[3]{abc} \implies \frac{s}{3} \geq \sqrt[3]{23} \implies s \geq 3\sqrt[3]{23}$$

This gives us a lower bound for $s$, but we need the exact value to minimize $a^2 + b^2 + c^2$. Maybe we can express $s$ in terms of $a$, $b$, and $c$ using the given constraints.

Let's consider $a$, $b$, and $c$ as the roots of the cubic equation:

$$t^3 - st^2 + 27t - 23 = 0$$

According to Vieta's formulas:

$$a + b + c = s, \quad ab + bc + ac = 27, \quad abc = 23$$

Now, we need to find $s$ such that this cubic equation has real positive roots. Perhaps we can find the value of $s$ that minimizes $a^2 + b^2 + c^2$.

From earlier, we have:

$$a^2 + b^2 + c^2 = s^2 - 54$$

To minimize $a^2 + b^2 + c^2$, we need to minimize $s^2$ (since $-54$ is constant). Therefore, we need to find the minimum value of $s$ such that the cubic equation has real positive roots.

Let's consider the cubic function:

$$f(t) = t^3 - st^2 + 27t - 23$$

For the cubic to have three real positive roots, certain conditions must be met. However, this approach might be too complicated. Perhaps we can consider specific values of $a$, $b$, and $c$ that satisfy the constraints.

Let's try assuming the box is a cube. If $a = b = c$, then:

$$2(ab + bc + ac) = 54 \implies 2(3a^2) = 54 \implies 6a^2 = 54 \implies a^2 = 9 \implies a = 3$$

But then the volume is:

$$abc = a^3 = 3^3 = 27$$

However, the volume is given as 23, not 27. So the box cannot be a cube.

Perhaps the box is close to a cube. Let's let $a = b$. Then the constraints become:

Surface area:

$$2(a^2 + 2ac) = 54 \implies 2a^2 + 4ac = 54 \implies a^2 + 2ac = 27 \implies c = \frac{27 - a^2}{2a}$$

Volume:

$$a^2 c = 23 \implies a^2 \left( \frac{27 - a^2}{2a} \right) = 23 \implies a \left( \frac{27 - a^2}{2} \right) = 23$$

Multiplying both sides by 2:

$$a(27 - a^2) = 46 \implies 27a - a^3 = 46 \implies a^3 - 27a + 46 = 0$$

Now, we need to solve the cubic equation $a^3 - 27a + 46 = 0$. Let's try to find rational roots using the Rational Root Theorem. The possible rational roots are factors of 46: $\pm 1, \pm 2, \pm 23, \pm 46$.
Trying $a = 2$:

$$2^3 - 27 \times 2 + 46 = 8 - 54 + 46 = 0$$

So $a = 2$ is a root. Therefore, we can factor the cubic:

$$(a - 2)(a^2 + 2a - 23) = 0$$

Setting each factor to zero:

1. $a - 2 = 0 \implies a = 2$ 2. $a^2 + 2a - 23 = 0$

Solving the quadratic equation:

$$a = \frac{-2 \pm \sqrt{(2)^2 - 4(1)(-23)}}{2} = \frac{-2 \pm \sqrt{4 + 92}}{2} = \frac{-2 \pm \sqrt{96}}{2} = \frac{-2 \pm 4\sqrt{6}}{2} = -1 \pm 2\sqrt{6}$$

Since dimensions must be positive, $a = -1 + 2\sqrt{6}$. However, let's first consider $a = 2$.

If $a = 2$, then:

$$c = \frac{27 - a^2}{2a} = \frac{27 - 4}{4} = \frac{23}{4}$$

Now, let's verify the volume:

$$a^2 c = 2^2 \times \frac{23}{4} = 4 \times \frac{23}{4} = 23$$

So $a = 2$, $b = 2$, $c = \frac{23}{4}$ satisfy both the surface area and volume constraints.

Now, let's calculate $a^2 + b^2 + c^2$:

$$a^2 + b^2 + c^2 = 2^2 + 2^2 + \left(\frac{23}{4}\right)^2 = 4 + 4 + \frac{529}{16} = 8 + \frac{529}{16} = \frac{128}{16} + \frac{529}{16} = \frac{657}{16}$$

Therefore, the space diagonal squared is:

$$a^2 + b^2 + c^2 = \frac{657}{16}$$

So the radius of the sphere is:

$$r = \frac{\sqrt{a^2 + b^2 + c^2}}{2} = \frac{\sqrt{\frac{657}{16}}}{2} = \frac{\sqrt{657}}{8}$$

Now, let's simplify $\sqrt{657}$:

$$657 = 9 \times 73 \implies \sqrt{657} = \sqrt{9 \times 73} = 3\sqrt{73}$$

Therefore:

$$r = \frac{3\sqrt{73}}{8} \implies r^2 = \left(\frac{3\sqrt{73}}{8}\right)^2 = \frac{9 \times 73}{64} = \frac{657}{64}$$

Now, let's consider whether this is the minimal $r^2$. Could other dimensions yield a smaller $r^2$?

Perhaps we can consider the general case using the identity we derived earlier:

$$a^2 + b^2 + c^2 = s^2 - 54$$

To minimize $a^2 + b^2 + c^2$, we need to minimize $s^2$, which means minimizing $s$. From the $AM \geq GM$ inequality, we have:

$$s = a + b + c \geq 3\sqrt[3]{abc} = 3\sqrt[3]{23}$$

So the minimum possible value of $s$ is $3\sqrt[3]{23}$. Let's calculate $s^2$ in this case:

$$s^2 = \left(3\sqrt[3]{23}\right)^2 = 9\left(\sqrt[3]{23}\right)^2 = 9 \times 23^{2/3}$$

Then:

$$a^2 + b^2 + c^2 = s^2 - 54 = 9 \times 23^{2/3} - 54$$

But this is not necessarily an integer, and it's not clear if this yields a smaller $r^2$ than the value we found earlier.

Perhaps the box with dimensions $a = 2$, $b = 2$, $c = \frac{23}{4}$ gives us the minimal $r^2$.

Let's consider another approach using Lagrange multipliers to minimize $a^2 + b^2 + c^2$ subject to the constraints $ab + bc + ac = 27$ and $abc = 23$.

Let's set up the Lagrangian:

$$L(a, b, c, \lambda, \mu) = a^2 + b^2 + c^2 - \lambda(ab + bc + ac - 27) - \mu(abc - 23)$$

Taking partial derivatives and setting them to zero:

1. $\frac{\partial L}{\partial a} = 2a - \lambda(b + c) - \mu bc = 0$ 2. $\frac{\partial L}{\partial b} = 2b - \lambda(a + c) - \mu ac = 0$ 3. $\frac{\partial L}{\partial c} = 2c - \lambda(a + b) - \mu ab = 0$ 4. $\frac{\partial L}{\partial \lambda} = ab + bc + ac - 27 = 0$ 5. $\frac{\partial L}{\partial \mu} = abc - 23 = 0$

From equations (1), (2), and (3):

$$2a - \lambda(b + c) - \mu bc = 0$$

$$2b - \lambda(a + c) - \mu ac = 0$$

$$2c - \lambda(a + b) - \mu ab = 0$$

Subtracting equation (1) from equation (2):

$$(2b - 2a) - \lambda(a + c - b - c) - \mu(ac - bc) = 0$$

$$2(b - a) - \lambda(a - b) - \mu(a - b)c = 0$$

Simplifying:

$$2(b - a) + \lambda(b - a) - \mu(b - a)c = 0$$

$$(b - a)\left[2 + \lambda - \mu c\right] = 0$$

So either $b = a$ or $2 + \lambda - \mu c = 0$.

Similarly, subtracting equation (2) from equation (3):

$$(2c - 2b) - \lambda(a + b - a - c) - \mu(ab - ac) = 0$$

$$2(c - b) - \lambda(b - c) - \mu(b - c)a = 0$$

Simplifying:

$$2(c - b) + \lambda(c - b) - \mu(c - b)a = 0$$

$$(c - b)\left[2 + \lambda - \mu a\right] = 0$$

So either $c = b$ or $2 + \lambda - \mu a = 0$.

Similarly, subtracting equation (3) from equation (1):

$$(2a - 2c) - \lambda(b + c - a - b) - \mu(bc - ab) = 0$$

$$2(a - c) - \lambda(c - a) - \mu(c - a)b = 0$$

Simplifying:

$$2(a - c) + \lambda(a - c) - \mu(a - c)b = 0$$

$$(a - c)\left[2 + \lambda - \mu b\right] = 0$$

So either $a = c$ or $2 + \lambda - \mu b = 0$.

From these equations, we have several possibilities:

1. $a = b = c$ 2. $a = b$, $2 + \lambda - \mu c = 0$, $2 + \lambda - \mu a = 0 \implies \mu c = \mu a \implies c = a$ 3. Similar cases leading to $a = b = c$

Therefore, the only solution is $a = b = c$.

But earlier, we found that $a = b = c$ leads to $a^3 = 27 \implies a = 3$, but the volume would be 27, not 23. So $a = b = c$ is not possible.

Therefore, the minimal $r^2$ occurs when $a = b$.

Thus, the dimensions $a = 2$, $b = 2$, $c = \frac{23}{4}$ give us the minimal $r^2$.

Therefore, $r^2 = \frac{657}{64}$.

Finally, we need to find $p + q$ where $r^2 = \frac{p}{q}$ and $p$ and $q$ are relatively prime positive integers.

Here, $p = 657$ and $q = 64$. Let's check if 657 and 64 are relatively prime.

The prime factorization of 64 is $2^6$.

The prime factorization of 657 is:

$$657 = 9 \times 73 = 3^2 \times 73$$

Since 64 and 657 have no common factors other than 1, they are relatively prime.

Therefore, $p + q = 657 + 64 = 721$.

Wait, let's make sure we've considered all possibilities. Perhaps there's a smaller $r^2$ with different dimensions.

Let's consider the case where $a$, $b$, and $c$ are all different.

From the earlier cubic equation:

$$t^3 - st^2 + 27t - 23 = 0$$

We need to find $s$ such that this cubic has three positive real roots.

Using the derivative of the cubic:

$$f'(t) = 3t^2 - 2st + 27$$

For the cubic to have three real roots, the discriminant of $f'(t)$ must be positive:

$$D = (-2s)^2 - 4 \times 3 \times 27 = 4s^2 - 324$$

Setting $D > 0$:

$$4s^2 - 324 > 0 \implies s^2 > 81 \implies s > 9 \text{ or } s < -9$$

Since $s = a + b + c$ and $a$, $b$, $c$ are positive, $s > 0$. Therefore, $s > 9$.

The minimum value of $s$ is 9, but $s$ must be greater than 9 for the cubic to have three real roots.

Earlier, we found that $s = a + b + c \geq 3\sqrt[3]{23} \approx 3 \times 2.8438 \approx 8.5314$.

Therefore, the minimum value of $s$ is approximately 8.5314, which is less than 9.

This suggests that the cubic may have three real roots when $s$ is slightly greater than 8.5314.

However, finding the exact value of $s$ that minimizes $a^2 + b^2 + c^2$ is complicated.

Therefore, the minimal $r^2$ occurs when $a = 2$, $b = 2$, $c = \frac{23}{4}$, giving $r^2 = \frac{657}{64}$.

Thus, the value of $p + q$ is $657 + 64 = 721$.

Let's verify that this is indeed the minimal $r^2$.

Suppose we try other values of $a$, $b$, and $c$ that satisfy the constraints.

Let's try $a = 1$.

From the volume constraint:

$$abc = 23 \implies bc = \frac{23}{a} = 23$$

From the surface area constraint:

$$2(ab + bc + ac) = 54 \implies ab + bc + ac = 27$$

Substituting $bc = 23$:

$$ab + 23 + ac = 27 \implies a(b + c) = 4$$

But $bc = 23$, so $b$ and $c$ are roots of the quadratic equation:

$$t^2 - (b + c)t + bc = 0 \implies t^2 - \left(\frac{4}{a}\right)t + 23 = 0$$

For $b$ and $c$ to be real, the discriminant must be non-negative:

$$\left(\frac{4}{a}\right)^2 - 4 \times 1 \times 23 \geq 0 \implies \frac{16}{a^2} - 92 \geq 0 \implies \frac{16 - 92a^2}{a^2} \geq 0$$

This implies:

$$16 - 92a^2 \geq 0 \implies 16 \geq 92a^2 \implies a^2 \leq \frac{16}{92} = \frac{4}{23} \implies a \leq \frac{2}{\sqrt{23}}$$

But $a = 1$ does not satisfy $a \leq \frac{2}{\sqrt{23}} \approx 0.4226$.
Therefore, $a$ cannot be 1.
This suggests that the minimal $r^2$ occurs when $a = 2$, $b = 2$, $c = \frac{23}{4}$.
Therefore, the value of $p + q$ is $657 + 64 = 721$.
As we've explored, the case where $a = b = 2$ and $c = \frac{23}{4}$ satisfies both the surface area and volume constraints and yields the minimal $r^2$. Any other dimensions either do not satisfy the constraints or result in a larger $r^2$.
A fleeting thought might suggest considering the box oriented differently within the sphere to potentially reduce $r$, but the minimal radius is inherently tied to the space diagonal of the box. Since the space diagonal is the longest distance between any two vertices, the sphere must have a radius of at least half the space diagonal to contain the box.
Another brief consideration might involve symmetry or other geometric properties, but the algebraic approach using the given constraints leads us directly to the solution.
Thus, the value of $r^2$ is $\frac{657}{64}$, and therefore $p + q = 657 + 64 = 721$.
Answer:** $\boxed{721}$

---

### A full example from MATH-500 Question-15 solved by T1

**Question**
Find the product of the $y$-coordinates of all the distinct solutions $(x, y)$ for the two equations $y = x^2 - 8$ and $y^2 = -5x + 44$.

**Response from T1**
Let's find the product of the $y$-coordinates of all the distinct solutions $(x, y)$ for the equations $y = x^2 - 8$ and $y^2 = -5x + 44$.
First, we'll express $x$ in terms of $y$ from the first equation:

$$y = x^2 - 8 \implies x^2 = y + 8 \implies x = \pm\sqrt{y + 8}$$

Now, let's substitute this expression for $x$ into the second equation $y^2 = -5x + 44$. Wait, but $x$ is in terms of $y$, so we'll have:

$$y^2 = -5(\pm\sqrt{y + 8}) + 44$$

This gives us two cases to consider due to the $\pm$:
**Case 1:** $y^2 = -5\sqrt{y + 8} + 44$
**Case 2:** $y^2 = 5\sqrt{y + 8} + 44$

Let's tackle Case 1 first:

$$y^2 = -5\sqrt{y+8} + 44$$

Rewriting it:

$$y^2 - 44 = -5\sqrt{y+8}$$

Multiplying both sides by $-1$ to make things neater:

$$44 - y^2 = 5\sqrt{y+8}$$

Now, let's square both sides to eliminate the square root:

$$(44 - y^2)^2 = \left(5\sqrt{y+8}\right)^2$$

Simplifying both sides:

$$(44 - y^2)^2 = 25(y+8)$$

Expanding the left side:

$$(44 - y^2)^2 = (44 - y^2)(44 - y^2) = 1936 - 88y^2 + y^4$$

So the equation becomes:

$$1936 - 88y^2 + y^4 = 25(y+8)$$

Expanding the right side:

$$1936 - 88y^2 + y^4 = 25y + 200$$

Bringing all terms to one side:

$$y^4 - 88y^2 - 25y + 1936 - 200 = 0$$

Simplifying:

$$y^4 - 88y^2 - 25y + 1736 = 0$$

Now, let's move on to Case 2:

$$y^2 = 5\sqrt{y+8} + 44$$

Rewriting:

$$y^2 - 44 = 5\sqrt{y+8}$$

Squaring both sides:

$$(y^2 - 44)^2 = \left(5\sqrt{y+8}\right)^2$$

Simplifying:

$$(y^2 - 44)^2 = 25(y+8)$$

Expanding the left side:

$$y^4 - 88y^2 + 1936 = 25(y+8)$$

Expanding the right side:

$$y^4 - 88y^2 + 1936 = 25y + 200$$

Bringing all terms to one side:

$$y^4 - 88y^2 - 25y + 1936 - 200 = 0$$

Simplifying:

$$y^4 - 88y^2 - 25y + 1736 = 0$$

Wait, that's the same quartic equation we got from Case 1! So both cases lead us to the same equation:

$$y^4 - 88y^2 - 25y + 1736 = 0$$

Now, we need to find the roots of this quartic equation. Let's try to factor it into two quadratics. Let's assume:

$$y^4 - 88y^2 - 25y + 1736 = (y^2 + ay + b)(y^2 + cy + d)$$

Expanding the right side:

$$(y^2 + ay + b)(y^2 + cy + d) = y^4 + (a + c)y^3 + (ac + b + d)y^2 + (ad + bc)y + bd$$
$$= y^4 + (a + c)y^3 + (ac + b + d)y^2 + (ad + bc)y + bd$$

Setting this equal to the original quartic:

$$y^4 + (a + c)y^3 + (ac + b + d)y^2 + (ad + bc)y + bd = y^4 - 88y^2 - 25y + 1736$$

Now, let's match the coefficients:
1. Coefficient of $y^3$:

$$a + c = 0 \implies c = -a$$

2. Coefficient of $y^2$:

$$ac + b + d = -88$$

Substituting $c = -a$:

$$a(-a) + b + d = -88 \implies -a^2 + b + d = -88$$

3. Coefficient of $y$:

$$ad + bc = -25$$

Substituting $c = -a$:

$$ad + b(-a) = -25 \implies ad - ab = -25 \implies a(d - b) = -25$$

4. Constant term:

$$bd = 1736$$

Now we have:
- $-a^2 + b + d = -88$ - $a(d - b) = -25$ - $bd = 1736$
Let's solve these equations. From the second equation:

$$a(d - b) = -25 \implies d - b = -\frac{25}{a}$$

So:

$$d = b - \frac{25}{a}$$

Now, substitute $d$ into the first equation:

$$-a^2 + b + \left(b - \frac{25}{a}\right) = -88$$

Simplifying:

$$-a^2 + 2b - \frac{25}{a} = -88$$

Multiply both sides by $a$ to eliminate the fraction:

$$-a^3 + 2ab - 25 = -88a$$

Bringing all terms to one side:

$$-a^3 + 2ab + 88a - 25 = 0$$

Rewriting:

$$-a^3 + 88a + 2ab - 25 = 0$$

This is getting messy. Perhaps there's a better approach. Let's try finding rational roots using the Rational Root Theorem.

The Rational Root Theorem tells us that any rational root $\frac{p}{q}$ of the polynomial $y^4 - 88y^2 - 25y + 1736$ has $p$ dividing the constant term 1736 and $q$ dividing the leading coefficient 1. So possible rational roots are the factors of 1736. Let's factor 1736:

$$1736 = 2 \times 868 = 2 \times 2 \times 434 = 2 \times 2 \times 2 \times 217 = 2^3 \times 217$$

Now, $217 = 7 \times 31$, so:

$$1736 = 2^3 \times 7 \times 31$$

Therefore, the possible rational roots are $\pm 1, \pm 2, \pm 4, \pm 7, \pm 8, \pm 14, \pm 28,$
$\pm 31, \pm 56, \pm 62, \pm 124, \pm 217, \pm 248, \pm 434, \pm 868, \pm 1736$.
Testing $y = 7$:

$$\begin{aligned}
7^4 - 88 \times 7^2 - 25 \times 7 + 1736 &= 2401 - 88 \times 49 - 175 + 1736 \\
&= 2401 - 4312 - 175 + 1736 \\
&= (2401 + 1736) - (4312 + 175) \\
&= 4137 - 4487 \\
&= -350
\end{aligned}$$

So $y = 7$ is not a root.
Testing $y = -7$:

$$(-7)^4 - 88 \times (-7)^2 - 25 \times (-7) + 1736 = 2401 - 88 \times 49 + 175 + 1736$$
$$= 2401 - 4312 + 175 + 1736$$
$$= (2401 + 175 + 1736) - 4312$$
$$= 4312 - 4312$$
$$= 0$$

So $y = -7$ is a root!

Great, we've found one root. Now, let's factor $y + 7$ out of the quartic polynomial.

Using synthetic division to divide $y^4 - 88y^2 - 25y + 1736$ by $y + 7$:

$$
\begin{array}{r|rrrrr}
-7 & 1 & 0 & -88 & -25 & 1736 \\
   &   & -7 & 49 & 273 & -1736 \\
\hline
   & 1 & -7 & -39 & 248 & 0
\end{array}
$$

The quotient is $y^3 - 7y^2 - 39y + 248$.

Now, we need to factor $y^3 - 7y^2 - 39y + 248$. Let's look for rational roots again using the Rational Root Theorem. The possible rational roots are the factors of 248:

$$248 = 2 \times 124 = 2 \times 2 \times 62 = 2 \times 2 \times 2 \times 31 = 2^3 \times 31$$

So the possible rational roots are $\pm 1, \pm 2, \pm 4, \pm 8, \pm 31, \pm 62, \pm 124, \pm 248$.

Testing $y = 8$:

$$8^3 - 7 \times 8^2 - 39 \times 8 + 248 = 512 - 7 \times 64 - 312 + 248$$
$$= 512 - 448 - 312 + 248$$
$$= (512 + 248) - (448 + 312)$$
$$= 760 - 760$$
$$= 0$$

So $y = 8$ is a root.

Now, let's factor $y - 8$ out of $y^3 - 7y^2 - 39y + 248$ using synthetic division:

$$
\begin{array}{r|rrrr}
8 & 1 & -7 & -39 & 248 \\
  &   & 8 & 8 & -248 \\
\hline
  & 1 & 1 & -31 & 0
\end{array}
$$

The quotient is $y^2 + y - 31$.

So, the quartic polynomial factors as:

$$y^4 - 88y^2 - 25y + 1736 = (y + 7)(y - 8)(y^2 + y - 31)$$

Now, let's find the roots of the quadratic $y^2 + y - 31$:

$$y = \frac{-1 \pm \sqrt{1^2 - 4 \times 1 \times (-31)}}{2}$$
$$= \frac{-1 \pm \sqrt{1 + 124}}{2}$$
$$= \frac{-1 \pm \sqrt{125}}{2}$$
$$= \frac{-1 \pm 5\sqrt{5}}{2}$$

So the roots are $y = \frac{-1+5\sqrt{5}}{2}$ and $y = \frac{-1-5\sqrt{5}}{2}$.

Therefore, the solutions for $y$ are:

$$y = -7, \quad y = 8, \quad y = \frac{-1+5\sqrt{5}}{2}, \quad y = \frac{-1-5\sqrt{5}}{2}$$

Now, we need to verify that these $y$-values correspond to valid solutions $(x, y)$ by checking the original equations.

Starting with $y = -7$:

From $y = x^2 - 8$:

$$-7 = x^2 - 8 \implies x^2 = 1 \implies x = \pm 1$$

Now, let's check $y^2 = -5x + 44$:

For $x = 1$:

$$(-7)^2 = -5(1) + 44 \implies 49 = -5 + 44 \implies 49 = 39$$

This is not true.

For $x = -1$:

$$(-7)^2 = -5(-1) + 44 \implies 49 = 5 + 44 \implies 49 = 49$$

This is true.

So $y = -7$ corresponds to the solution $(x, y) = (-1, -7)$.

Next, $y = 8$:

From $y = x^2 - 8$:

$$8 = x^2 - 8 \implies x^2 = 16 \implies x = \pm 4$$

Now, check $y^2 = -5x + 44$:

For $x = 4$:

$$8^2 = -5(4) + 44 \implies 64 = -20 + 44 \implies 64 = 24$$

This is not true.

For $x = -4$:

$$8^2 = -5(-4) + 44 \implies 64 = 20 + 44 \implies 64 = 64$$

This is true.

So $y = 8$ corresponds to the solution $(x, y) = (-4, 8)$.

Now, let's consider $y = \frac{-1+5\sqrt{5}}{2}$:

From $y = x^2 - 8$:

$$x^2 = y + 8 = \frac{-1+5\sqrt{5}}{2} + 8 = \frac{-1+5\sqrt{5}+16}{2} = \frac{15+5\sqrt{5}}{2}$$

So:

$$x = \pm\sqrt{\frac{15+5\sqrt{5}}{2}}$$

Now, let's check $y^2 = -5x + 44$:

First, compute $y^2$:

$$y^2 = \left(\frac{-1 + 5\sqrt{5}}{2}\right)^2$$

$$= \frac{\left(-1 + 5\sqrt{5}\right)^2}{4}$$

$$= \frac{1 - 10\sqrt{5} + 25 \times 5}{4}$$

$$= \frac{1 - 10\sqrt{5} + 125}{4}$$

$$= \frac{126 - 10\sqrt{5}}{4}$$

$$= \frac{63 - 5\sqrt{5}}{2}$$

Now, compute $-5x + 44$:

For $x = \sqrt{\dfrac{15 + 5\sqrt{5}}{2}}$:

$$-5x + 44 = -5\left(\sqrt{\frac{15 + 5\sqrt{5}}{2}}\right) + 44$$

$$= -5\left(\sqrt{\frac{5(3 + \sqrt{5})}{2}}\right) + 44$$

$$= -5\left(\sqrt{\frac{5(3 + \sqrt{5})}{2}}\right) + 44$$

This is getting messy, and it's clear that verifying this algebraically is quite involved. However, since we obtained $y$ by solving the quartic equation derived from the original equations, and we've already found valid solutions for $y = -7$ and $y = 8$, we can reasonably conclude that $y = \dfrac{-1 + 5\sqrt{5}}{2}$ and $y = \dfrac{-1 - 5\sqrt{5}}{2}$ are also valid solutions.

Similarly, for $y = \dfrac{-1 - 5\sqrt{5}}{2}$:

From $y = x^2 - 8$:

$$x^2 = y + 8 = \frac{-1 - 5\sqrt{5}}{2} + 8 = \frac{-1 - 5\sqrt{5} + 16}{2} = \frac{15 - 5\sqrt{5}}{2}$$

So:

$$x = \pm\sqrt{\frac{15 - 5\sqrt{5}}{2}}$$

Again, checking $y^2 = -5x + 44$ would be algebraically intensive, but we can accept these as valid solutions based on our earlier work.

Now, let's find the product of all the distinct $y$-coordinates:

$$y_1 = -7, \quad y_2 = 8, \quad y_3 = \frac{-1 + 5\sqrt{5}}{2}, \quad y_4 = \frac{-1 - 5\sqrt{5}}{2}$$

First, let's multiply $y_3$ and $y_4$:

$$y_3 \times y_4 = \left( \frac{-1 + 5\sqrt{5}}{2} \right) \left( \frac{-1 - 5\sqrt{5}}{2} \right)$$

$$= \frac{\left(-1 + 5\sqrt{5}\right)\left(-1 - 5\sqrt{5}\right)}{4}$$

$$= \frac{(-1)^2 - \left(5\sqrt{5}\right)^2}{4}$$

$$= \frac{1 - 25 \times 5}{4}$$

$$= \frac{1 - 125}{4}$$

$$= \frac{-124}{4}$$

$$= -31$$

Now, multiply $y_1$, $y_2$, $y_3$, and $y_4$:

$$y_1 \times y_2 \times y_3 \times y_4 = (-7) \times 8 \times \left( \frac{-1 + 5\sqrt{5}}{2} \right) \times \left( \frac{-1 - 5\sqrt{5}}{2} \right)$$

$$= (-7) \times 8 \times \left( \frac{\left(-1 + 5\sqrt{5}\right)\left(-1 - 5\sqrt{5}\right)}{4} \right)$$

$$= (-7) \times 8 \times \left( \frac{-124}{4} \right)$$

$$= (-7) \times 8 \times (-31)$$

$$= (-7) \times 8 \times (-31)$$

$$= (-7) \times (-248)$$

$$= 1736$$

Alternatively, we can use Vieta's formulas on the quartic equation $y^4 - 88y^2 - 25y + 1736 = 0$. For a general quartic equation $y^4 + ay^3 + by^2 + cy + d = 0$, the product of the roots is $(-1)^4 d = d$.

In our equation, $d = 1736$, so the product of the roots is 1736.

Wait, that's interesting. Even though we had to verify the validity of each root, Vieta's formula confirms that the product of all the roots is indeed 1736.

However, perhaps we should double-check that all these $y$-values correspond to valid solutions. We've already verified $y = -7$ and $y = 8$. Let's briefly check the other two.

For $y = \dfrac{-1 + 5\sqrt{5}}{2}$:

From $y = x^2 - 8$:

$$x^2 = y + 8 = \frac{-1 + 5\sqrt{5}}{2} + 8 = \frac{-1 + 5\sqrt{5} + 16}{2} = \frac{15 + 5\sqrt{5}}{2}$$

So:

$$x = \pm \sqrt{\frac{15 + 5\sqrt{5}}{2}}$$

Now, check $y^2 = -5x + 44$:

Compute $y^2$:

$$y^2 = \left(\frac{-1 + 5\sqrt{5}}{2}\right)^2$$

$$= \frac{\left(-1 + 5\sqrt{5}\right)^2}{4}$$

$$= \frac{1 - 10\sqrt{5} + 25 \times 5}{4}$$

$$= \frac{1 - 10\sqrt{5} + 125}{4}$$

$$= \frac{126 - 10\sqrt{5}}{4}$$

$$= \frac{63 - 5\sqrt{5}}{2}$$

Now, compute $-5x + 44$ for $x = \sqrt{\frac{15 + 5\sqrt{5}}{2}}$:

$$-5x + 44 = -5\left(\sqrt{\frac{15 + 5\sqrt{5}}{2}}\right) + 44$$

$$= -5\left(\sqrt{\frac{5(3 + \sqrt{5})}{2}}\right) + 44$$

$$= -5\left(\sqrt{\frac{5(3 + \sqrt{5})}{2}}\right) + 44$$

This is getting complicated, and perhaps it's time to accept that since these $y$-values satisfy the quartic equation derived from the original system, and we've verified the other roots, these are valid solutions.

Similarly, for $y = \frac{-1 - 5\sqrt{5}}{2}$, the calculations would be analogous.

Therefore, the product of the $y$-coordinates of all the distinct solutions is indeed 1736.

However, perhaps there's a more straightforward way to find the product without getting bogged down in the algebra.

Let's revisit the original equations:

1. $y = x^2 - 8$  2. $y^2 = -5x + 44$

From the first equation, $x^2 = y + 8$.

Substitute $x^2 = y + 8$ into the second equation:

$$y^2 = -5x + 44$$

But we still have $x$ in the equation. Perhaps we can express $x$ in terms of $y$ from the first equation:

$$x = \pm\sqrt{y + 8}$$

Substitute into the second equation:

$$y^2 = -5\left(\pm\sqrt{y + 8}\right) + 44$$

This gives us two cases:

**Case 1:** $y^2 = -5\sqrt{y + 8} + 44$

**Case 2:** $y^2 = 5\sqrt{y + 8} + 44$

Let's solve Case 1:

$$y^2 = -5\sqrt{y+8} + 44$$

Rewriting:

$$y^2 - 44 = -5\sqrt{y+8}$$

Multiply both sides by $-1$:

$$44 - y^2 = 5\sqrt{y+8}$$

Square both sides:

$$(44 - y^2)^2 = 25(y+8)$$

Expanding the left side:

$$(44 - y^2)^2 = 1936 - 88y^2 + y^4$$

So:

$$1936 - 88y^2 + y^4 = 25(y+8)$$

Expanding the right side:

$$1936 - 88y^2 + y^4 = 25y + 200$$

Bring all terms to one side:

$$y^4 - 88y^2 - 25y + 1936 - 200 = 0$$

Simplifying:

$$y^4 - 88y^2 - 25y + 1736 = 0$$

Now, Case 2:

$$y^2 = 5\sqrt{y+8} + 44$$

Rewriting:

$$y^2 - 44 = 5\sqrt{y+8}$$

Square both sides:

$$(y^2 - 44)^2 = 25(y+8)$$

This is the same equation as before:

$$y^4 - 88y^2 - 25y + 1736 = 0$$

So both cases lead to the same quartic equation.

Now, instead of factoring, perhaps we can use Vieta's formulas directly. For the quartic equation $y^4 + ay^3 + by^2 + cy + d = 0$, the product of the roots is $(-1)^4 d = d$.

In our equation $y^4 - 88y^2 - 25y + 1736 = 0$, the constant term $d = 1736$.

Therefore, the product of the roots is $1736$.

This confirms our earlier result.

As a final check, let's consider the geometric interpretation. The first equation $y = x^2 - 8$ is a parabola opening upwards, and the second equation $y^2 = -5x + 44$ can be rewritten as $x = \dfrac{44 - y^2}{5}$, which is a downward-opening parabola when plotted as $x$ vs. $y$.

Finding their intersections involves solving the quartic equation we derived, and we've found all the $y$-coordinates of the intersection points.

Therefore, the product of the $y$-coordinates of all the distinct solutions is indeed 1736.

While we initially tried to find the solutions by substituting and factoring, the calculations became quite involved. However, using Vieta's formulas provided a much more straightforward path to the solution.

So, after all that, the product of the $y$-coordinates is:

$$\boxed{1736}$$

