# OpenReview forum: "T1: Advancing Language Model Reasoning through Reinforcement Learning and Inference Scaling"
_ICML.cc/2025/Conference — ICML 2025 poster_

### Official Review · Reviewer_8Mb5 · 2025-03-11

**Overall Recommendation:** 2

**Summary:**

This paper examines the SFT + RL training pipeline for enhancing reasoning in LLMs. The authors propose several techniques to improve model performance, including SFT with critiques, auxiliary entropy bonuses, high-temperature sampling, Exponential Moving Average (EMA) stabilization, and On-policy KL normalization. These methods aim to refine reasoning quality, stabilize training, and optimize exploration in the RL framework.

**Claims And Evidence:**

The paper claims that SFT with critiques provides a strong foundation for RL training, but it lacks experimental evidence to support this assertion. Without concrete results, it is unclear how much critiques contribute to improving reasoning performance. Additionally, all proposed techniques—including SFT with critiques, auxiliary entropy bonuses, high-temperature sampling, EMA stabilization, and On-policy KL normalization—require a thorough ablation study to isolate their individual effects and validate their impact on model performance.

**Essential References Not Discussed:**

None

**Experimental Designs Or Analyses:**

The paper has serious issues in its experimental design. While multiple techniques are proposed, there is insufficient experimental validation to demonstrate the effectiveness of each method. A more rigorous evaluation, including ablation studies and comparative analysis, is necessary to substantiate the contributions of these techniques.

**Methods And Evaluation Criteria:**

While the proposed methods are conceptually sound, the paper lacks sufficient discussion on their effectiveness. There is minimal empirical analysis or justification for how each technique contributes to overall model improvement. A deeper exploration, supported by quantitative results and comparisons, is needed to validate their impact on reasoning performance.

**Other Comments Or Suggestions:**

None

**Other Strengths And Weaknesses:**

Strength
1.Explores supervised fine tuning (SFT) with critics + reinforcement learning (RL) as a means to enhance reasoning in LLMs.

Weaknesses
1.Lack of Empirical Validation – The paper proposes multiple techniques (e.g., SFT with critiques, auxiliary entropy bonus, high-temperature sampling, EMA, On-policy KL normalization) but lacks sufficient experimental results to support their effectiveness. An ablation study is needed.

2.Lack of detailed description of the proposed method.

**Questions For Authors:**

Lack of clarity on how critiques are used for SFT training:

1.If critiques are incorporated into SFT, what is the learning signal?

2.Is the model being explicitly trained to generate critiques alongside answers?

3.Is there a more powerful model used to generate critiques alongside answers?

4.What is the prompt used to generate the critiques?

5.How does the author ensure the correctness of the critiques?

Can the author provide an ablation study for the techniques proposed in the paper?

**Relation To Broader Scientific Literature:**

This paper has limited contributions in terms of the techniques it talks about.

**Theoretical Claims:**

This is not a theory paper.

---

> ### Author Rebuttal · Authors · 2025-03-31
>
> Thanks for your kind review and feedback!
>
> We would first like to emphasize that the primary contribution of this work is to propose a framework designed to effectively scale the reinforcement learning (RL) training of large language models (LLMs). This approach significantly enhances reasoning capabilities and offers a unique perspective on understanding inference scaling. Extensive experimental results on math and reasoning benchmarks clearly demonstrate the effectiveness of our method, and we hope you recognize both our contribution and the efficacy of our proposed framework.
>
> Additionally, we fully agree that comprehensive ablation studies are essential. To thoroughly validate each component's effectiveness, we have included extensive ablation analyses in our paper (see Tables 1, 2, 3, and Figure 2). For your convenience, we re-present these results here:
>
> 1. Effects of high temperature
> | Temperature | min-*p* | MATH500 | AIME | Omni-MATH-500 |
> |-------------|---------|---------|------|---------------|
> | 0.9         | 0       | 78.2    | 19.1 | 32.0          |
> | 1.1         | 0       | 84.6    | 29.0 | 37.8          |
> | 1.2         | 0       | 86.4    | 29.3 | 38.6          |
> | 1.3         | 0       | 84.6    | 24.3 | 36.4          |
> | 1.2         | 0.05    | 78.8    | 11.5 | 31.6          |
>
> 2. Effects of penalty design
>
> |              | Penalty | step40 | step80 | step120 | step160 |
> |--------------|---------|--------|--------|---------|---------|
> | OverLongRatio | ✔️       | 0%     | 2.6%   | 1.6%    | 0.7%    |
> |              | ✘       | 0%     | 4.1%   | 16.3%   | -       |
> | Accuracy(%)   | ✔️       | 78.6   | 80.1   | 81.2    | 81.2    |
> |              | ✘       | 79.0   | 79.2   | 76.4    | -       |
>
>
> 3. Effects of long CoT for SFT. It can be observed that RL with LongCoT (T1) significantly outperforms RL with short CoT (Qwen2.5-Instruct).
>
> | Model                       | MATH500 | AIME | Omni-MATH-500 |
>  |-----------------------------|---------|------|---------------|
> | (RL) Qwen2.5-14B-Instruct        | 78.9    | 13.7 | 30.1          |
> | (SFT) T1-SFT (Qwen2.5-14B)        | 77.2    | 10.3 | 28.5          |
> | (RL) T1 (Qwen2.5-14B)            | 87.4    | 30.5 | 38.6          |
> | -- | -- | -- | -- |
> | (RL) Qwen2.5-32B-Instruct        | 82.8    | 13.6 | 33.1          |
> | (SFT) T1-SFT (Qwen2.5-32B)        | 83.4    | 24.9 | 34.6          |
> | (RL) T1 (Qwen2.5-32B)            | 92.4    | 50.6 | 49.6          |
>
>
> For questions,
>
> Q1: If critiques are incorporated into SFT, what is the learning signal?
>
> - A1: In this work, we employ a prompt engineering pipeline (as detailed in Section 2.2.1) to generate responses accompanied by reflections. Critiques serve as intermediate outputs within this pipeline and are integrated into the final response rather than being presented separately. Consequently, our training employs traditional Supervised Fine-Tuning (SFT) with next-token prediction.
>
> Q2: Is the model explicitly trained to generate critiques alongside answers?
>
> - A2: No, critiques are inherently part of the generated response rather than explicitly trained as a separate output.
>
> Q3: Is there a more powerful model used to generate critiques alongside answers?
>
> - A3: We utilize Qwen-72B-Instruct and Gemini-1.5-Pro to initially generate Chains-of-Thought (CoT). Gemini-1.5-Pro is specifically employed to generate critiques. Finally, the o1-mini model (without access to its hidden CoT) integrates the generated CoT and critiques into the final response. Notably, except for o1-mini, these models demonstrate significantly poorer math reasoning performance compared to T1. Furthermore, since the o1-mini model's CoT is inaccessible, using responses from it without the explicit long CoT results in considerably weaker performance relative to T1.
>
> Q4: What is the prompt used to generate the critiques?
>
> - A4: The prompts used to generate the synthetic responses involve a pipeline of prompting with 3 prompts. We will add them into the appendix in the next version due to the space limit of rebuttal. To generate critiques, the key point is to ask the LLM to “ analyze what's wrong in every given wrong response and try to find the key point in every correct response”.
>
> Q5: How do the authors ensure the correctness of the critiques?
>
> - A5: We do not explicitly verify the correctness of critiques. Instead, we just expect the model to learn and initialize self-reflection behavior in the SFT stage and prioritize accuracy and effectiveness improvements during the Reinforcement Learning (RL) stage.

---

> > ### Comment · Reviewer_8Mb5 · 2025-04-04
> >
> > Thank you for the clarification and for sharing the ablation study experiment.
> >
> > Q1: How are critiques integrated into the final response? Are they simply concatenated?
> >
> > New question: It seems that a significant part of this project/paper’s contribution comes from its open-source nature. Could you elaborate on that?

---

> > > ### Author Response · Authors · 2025-04-04
> > >
> > > Thanks for your kind response!
> > >
> > > ### Q1: How are critiques integrated into the final response? Are they simply concatenated?
> > >
> > > A1: The critiques are not merely concatenated; instead, we employ an additional LLM to rewrite and integrate the attempts along with their critiques into a coherent final response. In our preliminary experiments, we initially attempted to directly concatenate all responses and critiques without further processing. However, we found that the resulting output was neither fluent nor effective. Consequently, we adopted an LLM-based integration approach.
> > >
> > > Specifically, as detailed in Section 2.2.1, we first concatenate multiple intermediate responses $({y_i}^N)$ and their associated critiques $({c_i}^N)$. This concatenated text is then provided to an LLM along with a "rewrite" instruction. The input prompt to the LLM is formatted as follows:
> > >
> > >
> > > ---
> > >
> > > prompt = """
> > >
> > > Show me a new solution to the given problem according to the reference solutions. The new solution should contain the meaningful failed attempts in the above answers, and you can learn from the mistakes and finally reach the correct results
> > > Keep all the above attempts to show you intelligent reasoning and reflection.
> > >
> > > You can do some cross-verification across different `correct responses` or use different methods to solve the problem and ensure correctness. You can use the failed attempts in the above answers to learn from the mistakes and finally reach the correct results.
> > >
> > > [Problem]
> > >
> > >
> > > [Response-1]
> > >
> > > [Critique-1]
> > >
> > > ...
> > >
> > > [Response-N]
> > >
> > > [Critique-N]
> > >
> > > """
> > >
> > > ---
> > >
> > > In this way, the LLM can gracefully integrate the attempts and critiques as a fluent final response with reasoning and reflection.
> > >
> > > ### Q2: It seems that a significant part of this project/paper’s contribution comes from its open-source nature. Could you elaborate on that?
> > >
> > > A2: All experiments in our paper are conducted using open-source models (Qwen2.5 and GLM-4-9B). While the original training data is publicly accessible, we have performed additional cleaning and curation.
> > > We will release our trained models, curated datasets for both supervised fine-tuning (SFT) and reinforcement learning (RL), as well as comprehensive training details, to facilitate further research and advancement within the open-source community.
> > >
> > > We hope that our response can help address your concerns. Should you have any additional questions or require further clarification, we would be pleased to discuss them further.

---

### Official Review · Reviewer_KYsA · 2025-03-13

**Overall Recommendation:** 4

**Summary:**

The paper proposes an RL based method to improve the reasoning / inference scaling capabilities of LLMs.

The paper discussed the importance of exploration and proposed an RL objective incorporating an entropy bonus to encourage diverse sampling. The paper also discusses specific tricks, such as format penalty, over-long length penalty, the size of rollout, the sampling temperature, and the min-p, and their impact on RL training.

The experiment showing the correlation between generation length / accuracy are very interesting.

It shows results on scaling inference thinking steps, where SFT model has a flat curve, but RL enables model to more effectively use the inference-scaling budget and improve on MATH, AIME, etc. It also proposes a hypothesis that not just correct steps matter, but wrong steps that led to correct solution also matters.

Overall, the paper presents very strong RL results on widely used benchmarks; its empirical analysis are clear and thought-provoking. It proposes novel thesis, and ways to understand RL reasoning training and inference scaling.


## update after rebuttal

My position still stand as accept.

**Claims And Evidence:**

Claim 1:
Exploration matters for RL learning and RL stability.

It shows empirical evidence that higher temperature in RL sampling, 1.1 and 1.2 gives better training result in Table-2.
Also, a greater rollout budget gives significantly better training result as in Figure-4.
However, what I failed to find is an ablation on the entropy penalty in the loss function, which seems to be a core claim of paper.


Claim 2:
Inference scaling is key and closely connected to RL reasoning learning.

The paper proposes a novel way to measure inference scaling by the truncating the thinking budget.
Figure-7 shows a consistent pattern that for RL models, having longer thinking budget improves reasoning accuracy. And truncating thinking process leads to performance loss for reasoning models.

**Essential References Not Discussed:**

Ablation on the token entropy loss seems to be missing and its impact not discussed.
Effect of the importance of CoT initialization is missing.

**Experimental Designs Or Analyses:**

Yes, i checked the main paper result, Table-1. To obtain stable numbers for small test set such as AIME, it averages the result across many runs.

**Methods And Evaluation Criteria:**

The proposed method and evaluation criteria are widely used and accepted, and makes sense.

**Other Comments Or Suggestions:**

NA

**Other Strengths And Weaknesses:**

Other Strength:
1. Proposes to analyze the inference-scaling effect in LLMs by truncating the reasoning trace, and using a summarization model to give final response. It unveils convincing patterns that reasoning models benefit from longer thinking budget, and is well-connected to inference time scaling.

**Questions For Authors:**

Have you tried other RL algorithms, eg. GRPO, and is the improvement you obtained specific to the algorithm being used.

Is the warm-start needed? The paper claims one distinction it makes is by considering both the correct and wrong steps of reasoning, but there don't seem to be enough evidence backing the claim.

Also, having a cold start baseline would be helpful to illustrate the importance of your proposed CoT initialization strategy. And if cold-start doesn't work, may be the correct response lie in a region difficult to reach from the base model, it will further illustrate the strength of the warm-up strategy the authors proposed.


Is there a plan to release the training code/setup?

**Relation To Broader Scientific Literature:**

It relates to the Warm-Start RL training in the DeepSeek-R1 technical report.

**Theoretical Claims:**

Yes, the claims of adding entropy penalty to RL loss to encourage exploration makes sense.
The connection it draws between inference time scaling and reasoning training makes sense.

---

> ### Author Rebuttal · Authors · 2025-03-31
>
> Thanks for your valuable feedback!
>
> ### Q1: Have you tried other RL algorithms ?
>
> We also conduct experiments using GRPO, and run the experiments using Qwen-14b with k=16 for efficiency. The results are shown as follows
>
> |     | AIME | Omni-math-500 | MATH500 |
> | --- | --- | --- | --- |
> | T1 w/ RLOO | 23.8 | 36.8 | 83.6 |
> | T1 w/ GRPO | 23.4 | 37.4 | 83.8 |
>
> It can be observed that T1 with RLOO and T1 with GRPO show similar performance, which demonstrates that the proposed techniques can bring consistent improvement based on all existing RL algorithms, including both RLOO and GRPO.
>
>
> ### Q2: Is the warm-start needed?
>
> A2: Yes, a warm-start is important for reinforcement learning (RL) training as it provides a strong initial foundation for reasoning patterns and encourages exploration. In comparison, for example, Qwen2.5-14b-Instruct, which is trained with RL but utilizes short-CoT, significantly underperforms compared to T1, an RL-trained model initialized from a LongCoT SFT model.
> Recent studies, such as DeepSeek-R1-Zero, have shown that RL training from a base model can also yield strong reasoning capabilities and self-reflection patterns. However, training directly from a base model typically requires an extensive training period to develop these cognitive patterns. A warm-start approach positions the model effectively from the outset, allowing continuous optimization and refinement of its reasoning abilities.
>
>
> | Model                       | MATH500 | AIME | Omni-MATH-500 |
> | ----------------------------- | --------- | ------ | --------------- |
> | (RL) Qwen2.5-14B-Instruct        | 78.9    | 13.7 | 30.1          |
> | (SFT) T1-SFT (Qwen2.5-14B)        | 77.2    | 10.3 | 28.5          |
> | (RL) T1 (Qwen2.5-14B)            | 87.4    | 30.5 | 38.6          |
> | -- | -- | -- | -- |
> | (RL) Qwen2.5-32B-Instruct        | 82.8    | 13.6 | 33.1          |
> | (SFT) T1-SFT (Qwen2.5-32B)        | 83.4    | 24.9 | 34.6          |
> | (RL) T1 (Qwen2.5-32B)            | 92.4    | 50.6 | 49.6          |

---

### Official Review · Reviewer_a8EM · 2025-03-17

**Overall Recommendation:** 3

**Summary:**

The paper introduces T1, a novel approach for enhancing the reasoning abilities of large language models (LLMs) by scaling reinforcement learning (RL) and leveraging inference compute. The method begins by initializing the LLM with synthesized chain‐of‐thought data that incorporates trial‐and‐error and self-verification, enriching the model’s reasoning patterns beyond simple correct-step replication. During subsequent RL training, T1 encourages extensive exploration by oversampling diverse responses using high-temperature sampling and incorporates a token-level entropy bonus along with on-policy KL normalization to stabilize training. These techniques boost the model’s ability to generate longer, coherent “thinking” sequences, and unlock an inference scaling property—where increased generation length directly correlates with improved reasoning performance.

Empirical evaluations are mainly conducted in math reasoning benchmarks (including AIME2024, MATH500, Omni-MATH-500, and GPQA), demonstrating that T1 outperforms both its supervised fine-tuning baseline and other state-of-the-art models including o1-preview and QwQ-32B-preview. The paper also proposes a simple strategy to measure inference scaling by truncating the generated reasoning process and showing that longer reasoning leads to more accurate final answers.

## update after rebuttal

My review still leans toward acceptance, but it could also be rejected. The major concern about "All training and most evaluations focus on math word problems and quantitative reasoning" is partially addressed with a few additional experiments (on one additional logical reasoning task), and there are some interesting findings like "our experiments indicate that training on mathematical problems generalizes effectively to other domains." I hope to see more insights into why and how math reasoning can be generalized to other domains (with more thorough empirical and, if possible, theoretical results). The claim of the next version should be re-defined in the center of math reasoning to avoid any confusion or misinterpretation from audiences.

**Claims And Evidence:**

The paper makes several claims that are largely supported by **empirical evidence**, though a few appear slightly overstated. The primary claim is that the proposed T1 model achieves superior performance on challenging math reasoning benchmarks and exhibits inference scaling behavior. This is backed by Table 1 in the paper, which shows T1 (with RL training) outperforming baseline models on multiple math problem datasets.

Another core claim is that T1 demonstrates inference scaling, meaning that allowing the model to “think” longer (increased inference budget) directly yields higher accuracy without external verification. The authors support this by conducting experiments in which they systematically truncate the chain of thought at varying lengths and measure performance.

The claim that T1’s exploration-oriented RL training yields better reasoning is supported by ablations, such as "Sampling more responses encourages exploration," "High temperature in sampling benefits RL training," and "Effects of penalty."

A minor overstatement: the claim that "T1 achieves superior performance across all benchmarks" is inaccurate—on GPQA, a baseline model slightly outperformed T1.

**Essential References Not Discussed:**

This paper addresses one of the most urgent research problems in training thinking models right now. Considering the short period of time this area has taken to raise, the important missing references are mostly the other concurrent works on improving LLM reasoning with RL, for example, DeepSeek's R1, s1, or any other claimed successful replications of o1 or R1. While it's unfair to ask this paper to compare comprehensively with others, we should expect a more thorough discussion should this paper go into ICML.

**Experimental Designs Or Analyses:**

Yes, the paper provides robust, well-executed experiments with detailed ablations, for example:
- comparisons with strong baselines (GPT-4, Claude, QwQ-32B) confirm T1’s effectiveness (Table 1)
- ablation studies validate key components: Scaling K improves exploration, while penalties prevent training collapse (Figures 3, Table 3)
- inference scaling is well-tested: Truncation experiments confirm that longer reasoning systematically improves accuracy

**Methods And Evaluation Criteria:**

Yes, the paper proposes an appropriate methodology and benchmarks for reasoning tasks.
- well-chosen benchmarks (AIME2024, MATH500, OmniMath, GPQA) objectively test math complex reasoning
- key design choices are justified, including synthetic chain-of-thought data, high K sampling (K=64) for RL, and entropy-based exploration

One major limitation is that the paper focuses heavily on math, training on math, and testing on math, thus limiting claims about general reasoning improvement.

**Other Comments Or Suggestions:**

N/A

**Other Strengths And Weaknesses:**

Strengths:
1. A major strength of the paper is how it creatively integrates several strategies – synthetic data generation, large-scale RL training, entropy regularization, etc. – into a cohesive framework. Each component (trial-and-error CoT pre-training, oversampling, penalty) is not entirely new on its own, but the way they are combined to achieve empirical improvements.
2. The paper’s clarity is generally strong. It is well-structured, first motivating the problem, then describing the T1 approach, and then providing extensive evaluation. The authors also clearly articulate the intuition behind each design choice (e.g., why high-temperature sampling is used and why they expect trial-and-error data to help). The writing is mostly easy to follow, and important points are supported by either references or experimental evidence.
3. The paper mentions that “The model weights and the data for SFT and RL training will be publicly available. If the authors follow through, this is a strong point of openness. It would allow the community to reproduce and build on T1. Considering the computational intensity of this approach, releasing the trained model and the large synthetic dataset would be highly valuable for further research.

Weaknesses:
1. A notable limitation is that all training and most evaluations are centered on math word problems and quantitative reasoning. While the paper argues this is a generic “reasoning” improvement, it is possible that some of the gains are specialized in math problem-solving. The trial-and-error CoT data is constructed for math questions; the reward function is based on checking numeric or symbolic answers. The authors do test one general puzzle dataset (GPQA) and see improvements​, which is encouraging, but a broader evaluation would strengthen the claims of general reasoning enhancement. In short, generality is a bit under-tested – the method’s success in math might be partly due to the structured nature of math problems (clear correctness criteria, availability of many similar problems for training). Future work could explore applying T1 to other reasoning benchmarks (e.g., Big-Bench reasoning tasks, logical deduction puzzles) to ensure the approach is universally beneficial.
2. Some critical training details are missing, including training infra config, compute cost, and more details on the training data. The approach is computationally heavy. RL with K=64 sampled responses per prompt, especially on a 32B model, implies a massive amount of GPU time. The paper doesn’t detail the compute used, but one can infer it is significant. This could be a practical weakness – the barrier to entry for others to apply this method is high. The authors mitigate this by planning to release the model, but training a new model with T1’s approach would be expensive. It’s worth noting that they chose relatively smaller base models (9B, 14B, 32B), likely because scaling to 70B or beyond with such an intensive training loop would be extremely costly. So, the method’s scalability in terms of engineering is a concern.

**Questions For Authors:**

Some of the questions are similar to the weaknesses or problems I raised above; the authors can combine them to reduce the overall response length.
1. The paper's evaluation primarily focuses on math benchmarks (MATH500, AIME, Omni-MATH), with a small test on GPQA. Could the authors provide clear results with meaningful baselines on non-mathematical reasoning tasks, such as multi-hop QA, commonsense reasoning, or logical deduction tasks?
2. Given that RL training was done with K=64 sampled responses per prompt, and models were trained up to 32B parameters, can the authors provide an estimate of GPU hours or FLOPs used for training?
3. The authors mention that the initial supervised fine-tuning (SFT) was done on synthetic reasoning data with trial-and-error and verification steps. Which model was used to generate this data (e.g., GPT-4, an earlier version of Qwen, etc.)?
4. The paper evaluates T1 at 9B, 14B, and 32B scales. Did the authors test whether similar RL training benefits models at a smaller scale (e.g., 7B or 3B)?
5. Did the authors observe high variance in results across different training runs? Does T1 always converge to a strong policy, or did some runs fail to improve significantly?
6. The authors demonstrate that RL improves reasoning over supervised fine-tuning (SFT) on synthetic data. How would T1 compare to an SFT model trained on a much larger dataset of human-annotated reasoning chains (e.g., thousands of expert solutions rather than synthetic ones)?

**Relation To Broader Scientific Literature:**

The paper studies the recent trends of using RL to improve LLM reasoning, i.e., test-time scaling. The paper’s contributions tie into the literature by confirming hypotheses that were hinted at in prior work (that RL and longer reasoning could significantly improve performance) and by introducing techniques (trial-and-error data augmentation, high-volume sampling in RL) that advance the understanding of how RL works.

**Theoretical Claims:**

There are no theoretical analyses in this paper.

---

> ### Author Rebuttal · Authors · 2025-03-31
>
> Thanks for your kind review and valuable suggestions!
>
> ### Q1 & W1: All training and most evaluations focus on math word problems and quantitative reasoning.
>
> Thank you for the question. We primarily conduct reinforcement learning (RL) training on math problems due to the following two reasons:
> Math problems can be reliably verified by rules, and thus the reward cannot be hacked during RL training.
> Math data are easily accessible in open-source communities.
> Data in other domains tends to be challenging either to verify reliably or to collect comprehensively. Nonetheless, our experiments indicate that training on mathematical problems generalizes effectively to other domains, and we actively evaluate our models in non-mathematical tasks as well, including GPQA and ZebraLogic. We found that training on math data can effectively lead to improvement in other reasoning tasks.
>
> | | GPQA | ZebraLogic |
> | -- | -- | -- |
> | T1-SFT (Qwen-14B) | 42.3 | 15.3 |
> | T1 (Qwen-14B) | 48.3 | 27.8 |
> | T1-SFT (Qwen-32B) | 49.5 | 20.3 |
> | T1 (Qwen-32B) | 56.1 | 27.9 |
>
>
>
> ### Q2 & W2: Critical training details, such as infrastructure configuration, compute costs, and detailed training data information, are missing.
>
> We employ 64 H800 GPUs, with each iteration taking approximately 300 seconds for a 14B parameter model (larger models require more GPUs and higher computational resources). Smaller K can lead to faster training. While training larger models involves substantial costs, smaller models also significantly benefit from RL training. Furthermore, ongoing improvements in training and inference infrastructure optimization within the open-source community are expected to continuously reduce the compute requirements for RL training and enhance overall training efficiency.
> For the training data, we use open-source data, including MATH-train, and NuminaMath . We split around 12k for the SFT stage and the others for RL training. And we will open-source the data later. You can find more training details in Appendix A.
>
> [1] MATH (train): Measuring Mathematical Problem Solving With the MATH Dataset
> [2] Numina-MATH: https://huggingface.co/datasets/AI-MO/NuminaMath-CoT/
>
> ### Q3: Could you provide an estimate of GPU hours or FLOPs used for training?
>
> We utilize 64 H800 GPUs, with each iteration requiring around 300 seconds for a 14B model. Since RL training combines generation and training processes, we have not recorded exact FLOP measurements.
>
> ### Q4: Which models are used for synthetic data generation?
>
> We leverage multiple models for synthetic data generation. Specifically, we use Qwen-72B-Instruct and Gemini-1.5-Pro to generate initial Chains-of-Thought (CoT), Gemini-1.5-Pro to generate critiques, and finally, the o1-mini model (without hidden CoT) to merge the CoT and critiques into the final answer. We experimented with several other models and discovered that different models excel in different aspects; thus, we utilize a combination of these models to produce high-quality final outputs.
>
> ### Q5: Have you conducted RL training on smaller models?
> A5: Currently, our experiments are mainly on 9B to 32B models and all of them shows significant performance improvement. We haven’t conducted RL training on smaller models. We think that we can also achieve observable improvement across different scale of models with RL.
>
>
> ### Q6: Is there significant variance across different RL training runs?
>
> In our experiments, we observed stable and consistent performance across multiple runs. Variance can be further reduced by increasing the sampling response (larger $K$). Overall, the improvements from RL training are consistent and reproducible across different runs.
>
> ### Q7: How would model T1 compare with a Supervised Fine-Tuning (SFT) model trained on a larger dataset of human-annotated reasoning chains?
>
> This is an insightful question and one that we are also exploring. Currently, nearly all available reasoning datasets are synthetically generated by models. Human-annotated datasets, such as GSM8k-train and MATH-train, usually contain short CoTs, leading to inferior performance. Open-source, high-quality, human-annotated reasoning datasets remain scarce, and it is unclear whether human-generated LongCoT data would outperform synthetic or RL-trained models. Annotating large-scale human reasoning data is challenging and costly, but we aim to investigate this further in future work.

---

### Official Review · Reviewer_7pmp · 2025-03-18

**Overall Recommendation:** 2

**Summary:**

This paper introduces T1, a method for enhancing LLM reasoning through reinforcement learning with increased exploration and inference scaling. The authors initialize a policy with chain-of-thought data incorporating trial-and-error patterns, promote exploration during RL through response oversampling, and analyze inference scaling by truncating reasoning steps.

**Claims And Evidence:**

The paper's empirical results show performance improvements on math reasoning benchmarks. However, the lack of comparison with other RL methods for LLMs (particularly GRPO) significantly undermines the credibility of the advancement claims. Without these comparisons, it's impossible to determine if T1 represents genuine progress over existing approaches.

**Essential References Not Discussed:**

GRPO, PPO

**Experimental Designs Or Analyses:**

The experiments focus on benchmark performance but lack comparative analysis against state-of-the-art RL methods for LLMs. This omission is critical since the authors claim to advance LLM reasoning through RL. The ablation studies are informative but insufficient without positioning against relevant baselines like GRPO or PPO.

**Methods And Evaluation Criteria:**

The exploration-encouraging RL approach has merit, but its novelty is questionable. The core motivation of encouraging exploration during RL training for better reasoning is critically similar to GRPO's.

**Other Comments Or Suggestions:**

No, I don't have any other comments

**Other Strengths And Weaknesses:**

Strengths:

Novel approach to scaling RL for reasoning
Clear experimental demonstration of inference scaling
Strong empirical results on challenging benchmarks
Practical implementation on open-source models

Weaknesses:

Limited comparison to alternative RL algorithms
Heavy focus on math reasoning with less emphasis on other domains
Limited analysis of the computational costs of the approach
Some details about reward modeling are underspecified

**Questions For Authors:**

How does the computational cost of T1 compare to other RL approaches for LLMs?
The paper mentions sampling K=64 responses which seems computationally intensive.

**Relation To Broader Scientific Literature:**

The paper inadequately discusses its relationship to recent work on RL for LLMs. While it mentions techniques like RLOO, it fails to thoroughly compare with and differentiate from other RL methods, which shares remarkably similar goals and motivations.

**Theoretical Claims:**

I don't see any theoretical claims to verify.

---

> ### Author Rebuttal · Authors · 2025-03-31
>
> Thanks for your valuable feedback!
>
> ### W1: Lack of comparison with other RL methods for LLMs (particularly GRPO)
>
> A1: We appreciate the reviewer's concern regarding the comparison with other RL methods, particularly GRPO and PPO. However, our work, T1, is designed to be independent of any specific RL framework, including GRPO, PPO, and RLOO. The core contribution of T1 lies in developing techniques to scale RL for LLMs, which are agnostic to the underlying RL method. Specifically, the techniques we propose can be integrated with any RL framework to enhance the reasoning abilities of LLMs.
>
> We chose RLOO as the primary RL framework due to its demonstrated stability and the strong performance observed in prior work. Notably, we find that RLOO and GRPO show similar results in our preliminary tests, reinforcing the idea that the techniques we introduce can be applied to both methods. Furthermore, the main contribution of our paper lies in how we leverage RL scaling to improve reasoning, a key area where existing methods, including GRPO and PPO, have not demonstrated substantial improvement in comparison to our findings.
> We also conduct experiments using GRPO, and run the experiments using Qwen-14b with k=16 for efficiency. The results are shown as follows
>
> |     | AIME | Omni-math-500 | MATH500 |
> | --- | --- | --- | --- |
> | T1 w/ RLOO (this work) | 23.8 | 36.8 | 83.6 |
> | T1 w/ GRPO | 23.4 | 37.4 | 83.8 |
>
> It can be observed that T1 with RLOO and T1 with GRPO show similar performance, which demonstrates that the proposed techniques can bring consistent improvement based on all existing RL algorithms, including both RLOO and GRPO.
>
>
> ### W2: The contribution is similar to GRPO
>
> A2: While both T1 and GRPO aim to improve RL for language models, our approach fundamentally differs from GRPO by directly targeting exploration scaling rather than refining the RL algorithm itself.
>
> - GRPO focuses on decomposing and improving PPO (e.g., by removing the value network and using group-wise reward normalization), with little emphasis on boosting exploration.
> - In contrast, T1 introduces strategies to help RL scaling for reasoning, such as trial-and-error learning with self-verification through chain-of-thought data, along with entropy bonuses and dynamic KL regularization—to systematically explore a broader reasoning space.
>
> T1 offers orthogonal contributions to algorithms like GRPO or RLOO and provides new insights into how RL can directly enhance the reasoning capabilities of LLMs, including RL training scaling and inference scaling.
>
> ### W3: Computational cost
>
> A3: We employ 64 H800 GPUs for all training, with each iteration taking approximately 300 seconds for a 14B parameter model with K=64.  Larger models require higher computational resources and smaller K can lead to faster training. Current open-source improvements in training and inference infrastructure optimization are expected to continuously reduce the compute requirements for RL training and enhance overall training efficiency.
>
> ### W4: More details about reward modeling.
>
> A4: We use rule-based reward with the correctness of response as the reward, 1 for correct and 0 for wrong. We ask the model to put the final answer within box and then use LLM to judge whether the model’s response equals to the groundtruth. You can find more details in Appendix A.

---

### Decision · Program_Chairs · 2025-05-01

**Decision:**

Accept (poster)

**Comment:**

The paper introduces T1, a method to enhance the reasoning capabilities of large language models (LLMs) through reinforcement learning (RL) and inference scaling. The approach involves initializing the model with synthesized chain-of-thought data and promoting exploration during RL training. The paper claims superior performance on (current/competitive) math reasoning benchmarks and introduces a strategy to examine inference scaling.

(+) Empirical Results: The paper provides strong empirical results on challenging benchmarks, showing T1's effectiveness.
(-) Limited Domain Evaluation: The focus is heavily on math reasoning, with limited evaluation on other reasoning domains, which raises questions about the generality of the approach.
(-) Comparative Analysis: There is a lack of comparison with other RL methods for LLMs, such as GRPO and PPO, which undermines the claims of advancement.

* Reviewer 7pmp: Weak reject due to insufficient comparison with other RL methods and limited domain evaluation.
* Reviewer a8EM: Weak accept, appreciating the novel approach but noting the need for broader evaluation.
* Reviewer KYsA: Accept, highlighting the strong empirical results and novel insights into inference scaling.
* Reviewer 8Mb5: Weak reject, citing lack of empirical validation for proposed techniques and insufficient ablation studies.

Overall, I recommend acceptance into the ICML program due to a well-studied method with strong experimental results (within its size/category – but convincing enough) on math reasoning benchmarks.